# Metabolic adaptation pilots the differentiation of human hematopoietic cells

Laëtitia Racine[1,2,3,4] ⓘ, Romuald Parmentier[1,2,3,4] ⓘ, Shreyas Niphadkar[5], Julie Chhun[1,2,3,4], Jean-Alain Martignoles[1,3,4] ⓘ, François Delhommeau[1,3,4] ⓘ, Sunil Laxman[5] ⓘ, Andras Paldi[1,2,3,4] ⓘ

**A continuous supply of energy is an essential prerequisite for survival and represents the highest priority for the cell. We hypothesize that cell differentiation is a process of optimization of energy flow in a changing environment through phenotypic adaptation. The mechanistic basis of this hypothesis is provided by the established link between core energy metabolism and epigenetic covalent modifications of chromatin. This theory predicts that early metabolic perturbations impact subsequent differentiation. To test this, we induced transient metabolic perturbations in undifferentiated human hematopoietic cells using pharmacological inhibitors targeting key metabolic reactions. We recorded changes in chromatin structure and gene expression, as well as phenotypic alterations by single-cell ATAC and RNA sequencing, time-lapse microscopy, and flow cytometry. Our observations suggest that these metabolic perturbations are shortly followed by alterations in chromatin structure, leading to changes in gene expression. We also show that these transient fluctuations alter the differentiation potential of the cells.**

## Introduction

The structural and functional reorganization during cell differentiation is a metabolically demanding process. Many proteins and other macromolecules are replaced, and the cytoskeleton is reorganized leading to changes in the cell's shape and motion. The gene expression profile is modified with many new genes that are transcribed, whereas other genes become repressed. This change in gene expression is invariably preceded by substantial chromatin remodeling initiated by epigenetic modifications (1). Cell differentiation is also accompanied by a metabolic transition from a predominantly glycolytic regime to an oxidative one (2). As for any change in a cell, these modifications would be impossible without mobilizing important internal metabolic resources. Biosynthetic

reactions of mRNA, protein, and other macromolecular synthesis are run by the hydrolysis of the high-energy bond of ATP and other trinucleotide phosphates; others are driven by the transfer of electrons from reduced transporters such as NADH, NADPH, or FADH$_2$. The energy demand is expected to be particularly high in the case of chromatin remodeling required for changes in gene expression because epigenetic modification reactions that determine chromatin stability represent a substantial energetic cost (3). These covalent modifications require different high-energy substrates, particularly S-adenosyl methionine and acetyl-CoA, which serve as donors to provide methyl and acetyl groups for histone and DNA methylation and acetylation. NAD$^+$, FAD, and $\alpha$-ketoglutarate, among others, are substrates required for reversal reactions that remove these groups from DNA and histones. Because of the rapid turnover of these modifications (3, 4, 5), the demand for the high-energy substrates is permanent even if the overall epigenetic profile remains globally unchanged and may substantially increase during major structural changes that accompany cell differentiation. This dependency of epigenetic modifications on the key metabolites therefore links the chromatin state to the metabolic state of the cell. The intracellular metabolite abundance is determined by the carbon and energy flux through their respective pathways. In turn, the flux through a given pathway is determined by the availability of the initial substrates and electron donors, the terminal electron acceptors, and the thermodynamic gradient defined by them (6). Most carbon sources and terminal electron acceptors are derived from the environment. Maintaining a constant energy flux requires continuous adjustment of the metabolic network to the fluctuations of the substrate concentrations in the given environment. Paradoxically, the metabolic activity of the cells itself may contribute to local changes in substrate concentrations. Small fluctuations can be rapidly buffered by the highly interconnected network (7).

Nevertheless, buffering alone is insufficient to compensate for the stress induced by large fluctuations. Changes in gene expression may be necessary to facilitate the cell's reorganization of

[1]Sorbonne Université, INSERM, Centre de Recherche Saint-Antoine, CRSA, Paris, France  [2]Ecole Pratique des Hautes Etudes, PSL Research University, Paris, France  [3]AP-HP, SIRIC CURAMUS, Hôpital Saint-Antoine, Service d'Hématologie Biologique, Paris, France  [4]OPALE Carnot Institute, Paris, France  [5]Institute for Stem Cell Science and Regenerative Medicine (DBT-inStem), Bangalore, India

Correspondence: andras.paldi@ephe.psl.eu

metabolic fluxes and enable it to metabolize new substrates or to migrate to a new environment. Given the highly interconnected nature of the metabolic network, fluctuations in one of the pathways inevitably impact all the others. The more extensive the reorganization of fluxes, the greater the impact on the intracellular concentration of "sentinel metabolites" strategically positioned at the crossroads of major catabolic and anabolic pathways ([8]). Because of the link between the "sentinel metabolites" and epigenetic modifications, metabolic fluctuations can be conveyed to the chromatin by modifying the dynamic balance of epigenetic reactions. A key question is defining what can be considered as a "large" metabolic fluctuation. Although it is difficult to set a clear limit in terms of intracellular metabolite abundance, large perturbations can be detected based on their immediate phenotypic effects on the cell division rate and growth. These processes require additional energy and carbon resources and serve as a reliable proxy for tracking the overall metabolic state. Chromatin remodeling, a prerequisite for changing gene expression patterns and, consequently, cell differentiation, consumes a substantial portion of the cell's energy production ([3], [8]). Therefore, it is reasonable to hypothesize that metabolic stress and cell differentiation are not only coupled, but also the former triggers the latter. To assess our hypothesis, we used human cord blood–derived CD34[+] cells. Recent studies have shown that major changes occur during the first 24 to 96 h after the in vitro stimulation of these cells ([1], [9]). The earliest changes involve rapid and extensive non-specific chromatin opening, rendering most gene promoters in the genome accessible for transcription. Consequently, by the end of the first cell cycle, ~48 h later, cells enter a phase of quasi-random hyper-transcription. This state is referred to as the "multilineage primed state" because of the simultaneous expression of genes characteristic of concurrent cell lineages. It serves to prepare the cells to respond to a variety of environmental changes. In the hematopoietic cells used in our studies, this process spans several cell cycles and is characterized by strong morphological fluctuations ([9]). By the end of the first two–three cell cycles, between 72 and 96 h post-stimulation, distinct gene expression profiles start to emerge, indicating that the cells are about to complete the first stage of the fate decision process. It is also known that multipotent hematopoietic cells and lineage-committed progenitors display markedly distinct metabolic characteristics ([10], [11], [12], [13], [14], [15]). Does the change of the metabolic setup of the cells precede and trigger the non-specific chromatin opening? Does the nature of metabolic stress impact the cell fate decision process?

To test our hypothesis regarding the role that metabolic stress plays in initiating and driving cell differentiation, we partially and transiently inhibited different essential metabolic pathways using small molecular inhibitors. Subsequently, we evaluated the short- and long-term consequences on the proliferation and differentiation of the cells in vitro.

The cell's energy metabolism relies on three major energy and carbon sources: glucose, amino acids (primarily glutamine), and lipids ([16]). In the present study, we specifically investigated the roles of glucose and glutamine. The targeted metabolic pathways along with the inhibitors used are schematically represented in Fig S1. We used 2-deoxy-D-glucose (2-DG) to inhibit the first step of glucose use by glycolysis, thereby reducing all downstream

processes. This reduction of glycolysis diminishes ATP production and is expected to shift the balance between NADP[+] and NADPH toward the reduced form. Indeed, the pentose–phosphate pathway is the major source of NADPH, an essential electron donor in lipid biosynthesis. The contribution of glucose to the carbon flow through pyruvate and acetyl-CoA to lipid biosynthesis and the Krebs cycle is also expected to decrease.

The second inhibitor, 6-diazo-5-oxo-L-norleucine (DON), inhibits the conversion of glutamine to glutamate by glutaminase (GLS). Glutamine plays a very important role in cellular metabolism by providing carbon and nitrogen to directly fuel biosynthetic processes in the cytoplasm, such as the biosynthesis of nucleotides or glutathione. After import into the mitochondria through a glutamine transporter and deamination to glutamate by GLS, glutamine-derived carbon atoms enter the TCA cycle and are used either for ATP production by terminal oxidation or for biosynthetic processes. DON action is expected to perturb all these downstream processes. However, glutamine is a non-essential amino acid and can be synthesized by the cell. Therefore, the effects of DON are expected to be at least partially mitigated by the alternative amino acid sources.

To perturb the glutamine pathway in a more targeted way, we used aminooxyacetate (AOA) to inhibit the glutamate-pyruvate and glutamate-oxaloacetate aminotransferases ([17]). These enzymes are essential because they ensure the redistribution of nitrogen between the amino acids and produce α-ketoglutarate (often referred to as αKG), a sentinel metabolite and essential TCA cycle intermediate. In this way, glutamine supplies carbon for ATP production by terminal oxidation or for biosynthesis of sugars and lipids. It also serves as a substrate for histone and DNA demethylation, thereby directly impacting chromatin structure.

All three drugs are known to induce substantial perturbation of the cellular energy metabolism and interfere with differentiation of hematopoietic cells ([18]). Here, we investigate the immediate effects of metabolic perturbations induced by these inhibitors on the CD34[+] cells at the moment of cytokine stimulation, their consequences on the initial steps of cell fate decision, and the long-term impact on differentiation.

We used time-lapse microscopy, cytometry with a generation tracker, single-cell and bulk ATAC-seq, and single-cell mRNA sequencing. Short-term effects of these perturbations on the cell proliferation and morphology, dynamics of the chromatin structure, and gene expression were recorded. The potential long-term effects on differentiation capacity resulting from transitory inhibition were studied after in vitro culture.

## Results

The experimental strategy shown in Figs 1A and S1 was based on the hypothesis that early metabolic priming impacts the fate choice of the cells at later stages. First, we followed the cell's dynamic behavior and morphological changes under various metabolic conditions during the earliest stages after the cytokine stimulation at 0 h (see the Material and Methods section). We used targeted quantitative mass spectrometry to assess the general changes in

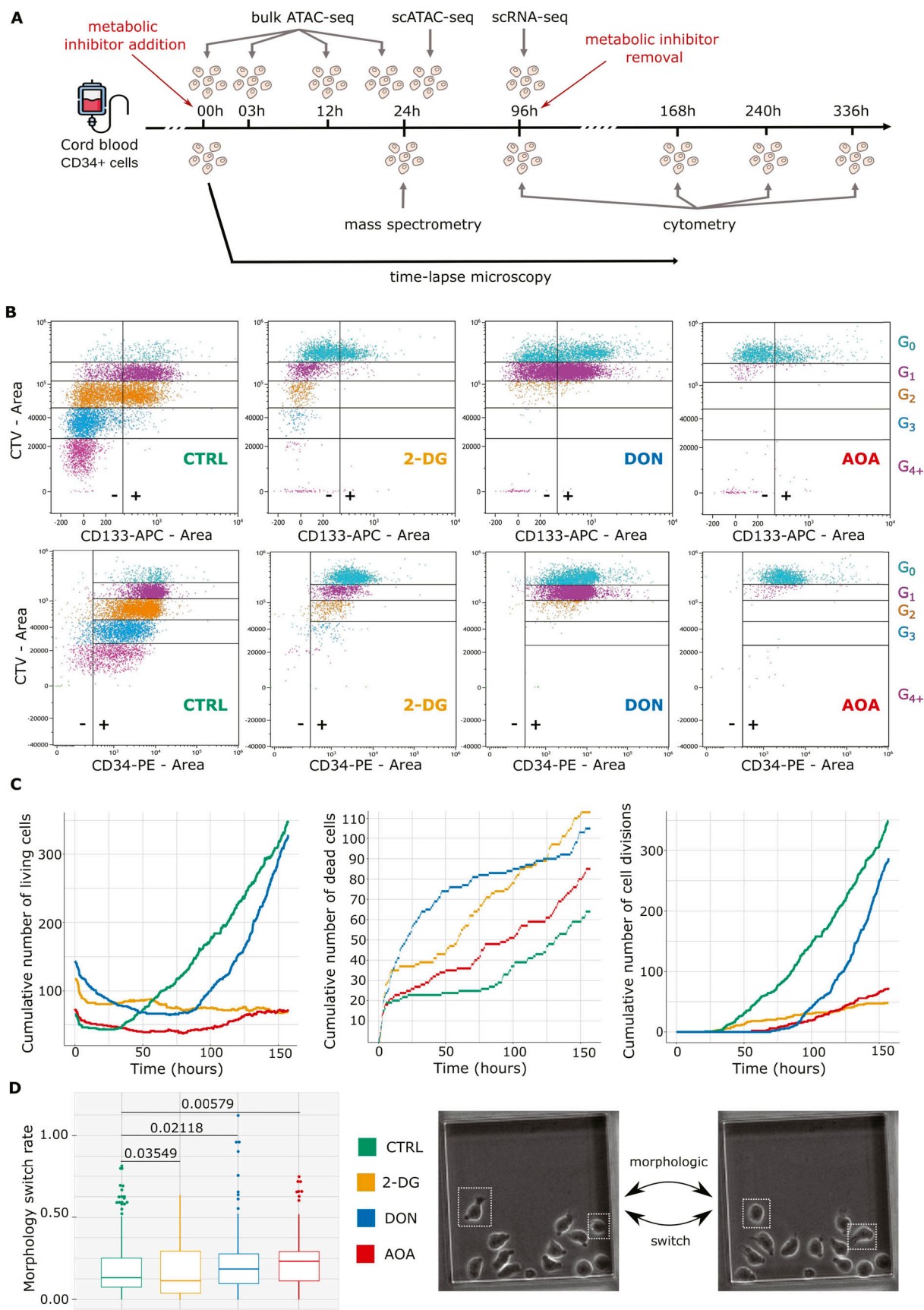

cell metabolism after inhibitor treatment. Subsequently, the early chromatin changes were identified using bulk and single-cell ATAC-seq at different time points during the first 24 h. To assess whether the cellular phenotype is under the influence of prior metabolic changes, the gene expression patterns were sampled at 96 h post-stimulation using single-cell RNA sequencing. Finally, the differentiation potential of the cells was assessed at later time points by flow cytometry using a set of cell surface markers. To do this, the cells were first cultured in the presence of inhibitors for a period of 4 d, then transferred to a new medium without inhibitors and a different cytokine composition (see the Material and Methods section) for an additional 10 d. Cell surface marker characterization was performed for each condition at days 4, 7, 10, and 14.

### Early metabolic perturbation strongly affects cell behavior

In a pilot study, we determined the sublethal concentrations that allowed survival of about half of the treated cells (1 mM 2-DG, 2.5 $\mu$M DON, and 1.9 mM AOA). We assume that at this concentration, the targeted pathways are only partially inhibited in the surviving cells. At 24 h, small molecular metabolites were extracted from ~$10^5$ cells after rapid quenching (see the Materials and Methods section) and analyzed by targeted quantitative mass spectrometry (19). The cumulative normalized results of nine experiments shown in Fig S2 indicate that the use of inhibitors induced substantial metabolic perturbations. We measured the relative amounts per condition of 11 metabolites: pyruvate, lactate, citrate, $\alpha$-ketoglutarate, malate, succinate as indicators of the glycolytic and the tricarboxylic acid cycle activity, and amounts of aspartate, tyrosine, glutamine, and the pair S-adenosylhomocysteine/S-adenosyl methionine. The relative amounts of all these metabolites were significantly altered compared with the control condition, and the responses were distinct depending on the enzyme inhibitor used. To make sense of these changes, we must consider the highly interconnected nature of the metabolic network and its complex behavior. Therefore, while targeting specific enzymes (Fig S1), the inhibitors are also expected to have wide-ranging pleiotropic effects, making it challenging to predict how individual metabolite abundance will change. Nevertheless, our measurements show significant alterations compared with the control, indicating real metabolic perturbations induced by the stress. Different inhibitors induced various alterations, and we also observed high sample-specific variations. Given the heterogeneous constitution of the CD34$^+$ cell population, we postulate that the cells developed multiple adaptation strategies

exhibiting a broad spectrum of responses for the diverse metabolic impairments induced by the inhibitors.

We characterized the cells using flow cytometry with Cell Trace Violet labeling to track the number of cell divisions, and the evolution of the multipotent cell markers CD34 and CD133 during the first 96 h after stimulation. A representative example is displayed in Fig 1B. Similar tendency was observed in all donors with some differences in kinetics (Fig S3). As described previously, the untreated control cells underwent one or two divisions, with only a small proportion of them dividing three or four times (9). The cells treated with 2-DG followed roughly the same scheme but divided much slower. Indeed, the proliferation index of 2-DG–treated cells was only 1.28 when untreated cell's one was 3.46 (Table S1). DON and AOA strongly inhibit cell divisions. Only half of the cells treated with DON made a single division, the second half remained undivided for 96 h, whereas cells treated with AOA did not divide at all (Fig 1B). As expected, the CD34 marker decreased at each cell division in all conditions. Overall, both the inhibition of glycolysis and glutaminolysis accelerated this process (Table S1). CD133 followed similar dynamics; however, its complete loss was more rapid with about half of the cells becoming negative after few divisions. CD133 decreased only slightly in DON-treated cells, whereas AOA-treated cells almost completely lost this marker even without division. The most significant impact is observed in the presence of 2-DG where the loss of the marker was reached after the first division.

To further characterize the dynamic behavior of the cells during the same 96-h period but also beyond this time window, we made time-lapse records of individual cells during a period of 6 d with a frequency of an image every 2 min (see the Material and Methods section). The cumulative counts of the number of living cells, cell deaths, and cell divisions are shown in Fig 1C. Representative videos are shown in Video 1. All conditions invariably exhibited a short initial period of high cell death rate typical to this type of experiments, generally attributed to the backlash of cell thawing. However, the cell survival at later stages was substantially different between each of the inhibitor-treated conditions. After a long 48-h first cell cycle, non-treated cells divided regularly with low death rate, and the population increased exponentially. Cells treated with 2-DG also started to divide after 48 h, but the rate of cell divisions remained low, and the death rate remained high. As a result, the population size remained largely unchanged during the entire recording period. In contrast, the cellular dynamics was very different in the case of DON and AOA. The first divisions of both DON- and AOA-treated cells were observed relatively late, occurring

**Figure 1. Experimental strategy and disturbances in cell behavior during early metabolic perturbation.**
**(A)** Experimental strategy: human cord blood–derived CD34$^+$ cells were cultured with or without one of the following metabolic inhibitors: DON 2.5 $\mu$M, AOA 1.9 mM, or 2-DG 1 mM. Mass spectrometry was performed on cells at 24 h to confirm the metabolic perturbation. The chromatin accessibility profile was analyzed by bulk ATAC-seq at 00, 03, 12, and 24 h and by single-cell ATAC-seq at 24 h. Cell's transcriptomes and phenotypes were analyzed by scRNA-seq and cytometry with generations tracking marker at 96 h. Continuous observations of the cells were performed during around 160 h using time-lapse microscopy. The long-term effects on differentiation were assessed by cytometry at 168, 240, and 336 h after inhibitor removal. **(B)** Membrane expression of CD133 and CD34 multipotent markers and CTV labeling across cell generations at 96 h. Results obtained from the cells of a representative donor are shown. Generations are indicated by the different colors. G0 is the parental generation. Positive and negative populations are delimited by a vertical line and −/+ signs. A decrease in mean fluorescence intensity can be detected by the population shift on the left on the x-axis. Other examples are shown in Fig S3. Detailed proportions of positive populations and mean fluorescence intensities per generation are shown in Table S2. **(C)** Cumulative number of living and dead cells and the total number of cell divisions during the first 160 h based on the time-lapse records. Each experimental condition is represented by a curve. Color codes are available under the graphs. **(D)** Switch rates of the cells depending on their culture condition as detected by time-lapse records. This rate is calculated by dividing the number of morphological switches by the cell cycle duration. The closer it is to 1, the more the cell has changed morphology during its cycle. P-values of pairwise comparisons using the Wilcoxon test are indicated. The two successive snapshots on the right panel show the morphological switch of two cells.

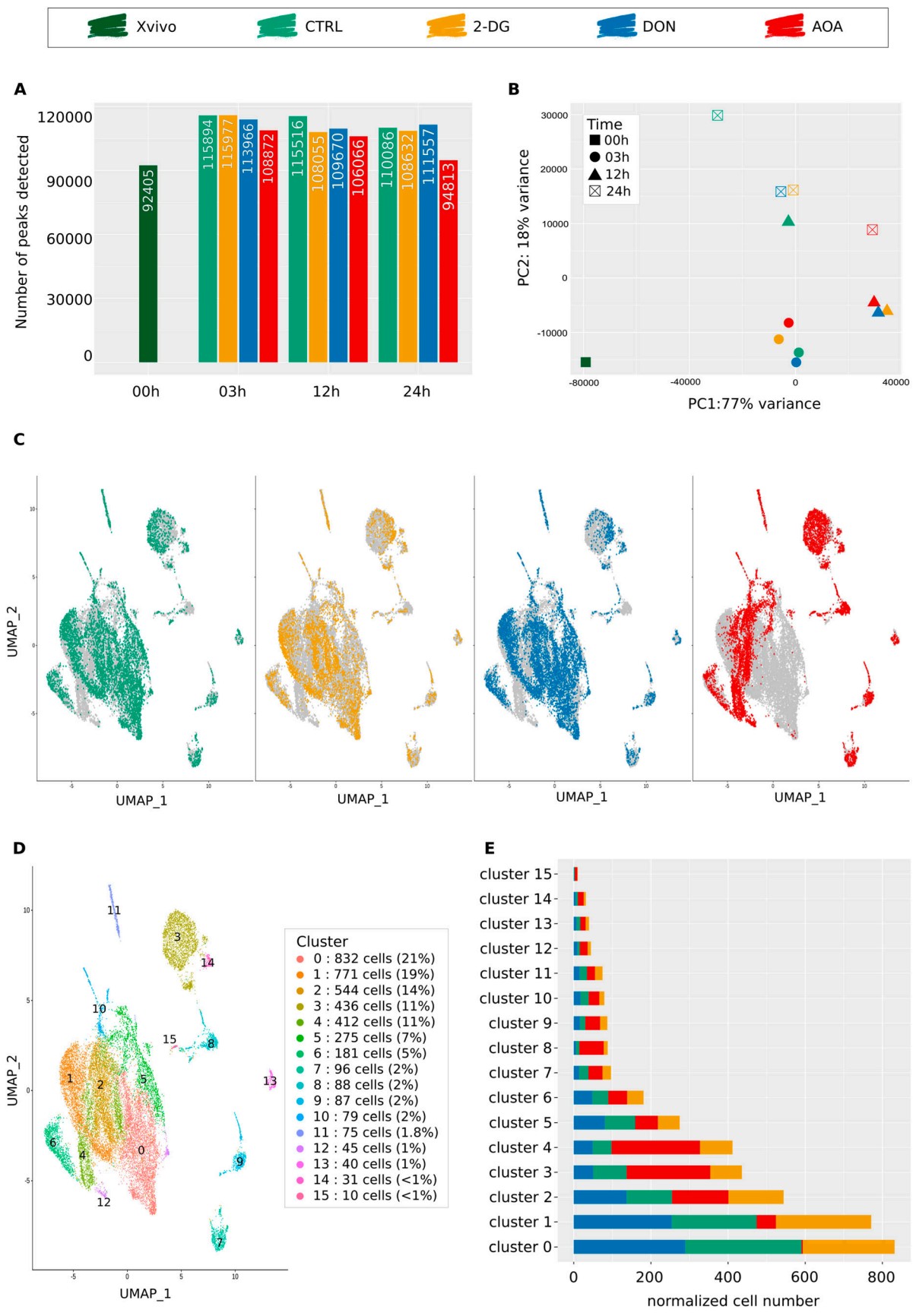

approximately between 70 and 90 h. The initial death rate of DON-treated cells was much higher than in the other conditions. However, it decreased after the first division, coinciding with an increase in the division rate. From this point onward, DON-treated cells exhibited a higher proliferative rate compared with the control group suggesting potential cellular adaptation to the metabolic stress. The death and division rate of the AOA-treated cells remained consistently high resulting in a population with a cumulatively low cell density.

Previous observations have shown that CD34[+] cells adopt two morphological phenotypes, either round or polarized. A large fraction of the cells were able to switch with a high frequency between the two forms. The stabilization in either of these phenotypes is associated with the emergence of distinct transcriptional profiles (9). Thus, we examined whether the metabolic stress influenced the fluctuating behavior of the cells. To do this, we tracked the cells on the time-lapse records and counted the number of morphological switches. Despite the significant heterogeneity between the clones, we observed an overall trend at the population level. As shown in Fig 1D, the switch frequency of AOA- and DON-treated cells increased slightly but significantly compared with the controls, although the frequency of the 2-DG-treated cells was slightly lower.

Taken together, the initial observations show that the inhibitors induce rapid metabolic perturbations within the first few hours after cell stimulation. These perturbations alter essential metabolite levels in the cells, impact the cell viability and division, and induce phenotypic changes. We are aware that critical chromatin changes are known to occur during the first 24 h (1). Therefore, we sought to assess how the chromatin structure responds to the metabolic perturbations and how these perturbations impact the gene expression.

## Initial metabolic stress impacts chromatin accessibility and gene expression

We investigated the genome-scale changes of the chromatin structure using whole-cell population-level ATAC-seq (referred to as bulk ATAC-seq) at 0, 3, 12, and 24 h after the stimulation of the cells. First, we recorded the number of accessible DNA regions (peaks) in each condition and at each time point (Fig 2A). As expected, based on our previous study, we observed a sharp increase in the number of peaks simultaneously with the stimulation of the cells. This increase, comparable to the control in all conditions, is the consequence of the global non-specific chromatin decompaction characterized earlier (1). Between 3 and 24 h, the number of

peaks remained stable or decreased slightly. The largest decrease, about 13%, was observed in AOA-treated cells, meaning that the chromatin remained slightly more compacted than in controls. A possible explanation could be the change in metabolic substrate availability required for chromatin opening.

Next, we analyzed how the size of common peaks between the control and each condition changed over time. As a proxy for the size of a peak, we used the number of sequenced reads (read counts) that define it. The increase or decrease in read counts for a peak in the same genomic position was interpreted as the likelihood of the chromatin to open or close. We calculated the log fold changes of the number of reads of each peak between conditions and the associated P-values. The pairwise comparisons at all time points are represented as volcano plots in Fig S4. Overall, the number of peaks with increased or decreased size was found equilibrated with the significant exception of AOA-treated cells at 24 h. Here, the number of peaks with decreased size compared with the control was strikingly high.

To establish whether the metabolic perturbations also induced modifications in the pattern of accessibility, we performed principal component analysis (PCA). The PCA plot in Fig 2B shows that both the control and inhibitor-treated cells undergo rapid and similar changes between 0 and 3 h. At 12 h, control cells are clearly separated from the other three conditions. 24 h after stimulation, AOA-treated cells are clearly separated from the two other conditions and map to the opposite position of the control untreated cells (Fig 2B). The analysis confirms that global rearrangement of chromatin after cell stimulation happens at the same rapid rate and to the same extent in all conditions, but the pattern of it starts to diverge soon after. In addition, it shows that the effect of the metabolic perturbations is clearly detected at 12 h and results in a substantially different pattern after 24 h. AOA-treated cells not only have the lowest number of accessible DNA regions, but also have a different pattern compared with the other two conditions or the control.

To further characterize these differences, we performed single-cell ATAC-seq on the cells at 24 h, when the changes were the most noticeable in the bulk analysis. In accordance with our previous study (1), the single-cell ATAC approach successfully identified the peaks also detected by the bulk version. The cumulative peak numbers confirmed the lower total number of peaks in AOA-treated cells compared with the other conditions. This results from the lower number of accessible DNA regions in individual cells (median count to 14,155 peaks per cell for AOA condition versus 16,973–18,913 peaks per cell for other conditions; see Table S2).

**Figure 2. Bulk and single-cell ATAC-seq analysis of chromatin accessibility at 24 h after the start of metabolic inhibitor's exposure.**
00-h Xvivo condition corresponds to cells cultured in a medium without early cytokines, before stimulation. **(A)** Total number of accessible regions (peaks) detected in all three independent donors at four different time points and in four culture conditions as determined by bulk ATAC-seq. Chromatin decompaction occurs in all conditions between 00 and 03 h, timing at which the highest number of peaks is reached. AOA-treated cells are the only one to display a noticeable recompaction at 24 h. **(B)** Principal component analysis based on differential accessibility analysis of bulk ATAC-seq data. Each point is defined by its color corresponding to its condition and its shape indicating the associated timing. **(C)** UMAP visualization of the scATAC-seq data at 24 h. Each dot represents a cell. The gray profile represents the shape of the total population when all conditions are merged. Color dots represent the cells of the global population corresponding to the studied condition (CTRL, 2-DG, DON, or AOA). **(D)** Cluster identification based on UMAP representation and the Louvain algorithm (resolution 0.25). A normalization step was performed before analyzing cell's distribution in clusters to consider the same total number of cells for each condition. All conditions were normalized to 1,000 cells. **(E)** Cell's distribution in scATAC-seq clusters depending on their culture condition.

To assess the heterogeneity of the chromatin response to metabolic perturbations, we visualized the cells on the same UMAP (Fig 2C). The four conditions showed different and heterogeneous profiles. Overall, 16 clusters were found and numbered from 0 to 15 (Fig 2D). The first six large clusters totalized around 90% of the cells, whereas the remaining 11 contained only a few percent each. All but one cluster contained cells from each condition, but the distribution of the control and treated cells was unequal between the different clusters (Fig 2E). This reveals that the different perturbations generated partially different patterns. This was particularly true for the AOA-treated cells. The largest cluster (n°0) contained no AOA-treated cells, whereas the control, 2-DG-, and DON-treated cells were represented at roughly equal proportions. The composition of cluster n°1 was very similar with only 6.5% of AOA-treated cells contrary to the ±30% of cells of the other conditions. In contrast, AOA-treated cells dominated the clusters 3, 4, and 8. Clusters n°2 and 5 were composed of roughly similar proportions of control and metabolically perturbed cells of the three types.

The single-cell ATAC-seq results confirmed that as suggested by the bulk ATAC-seq analysis, the different metabolic perturbations generated different chromatin responses. The response to each condition was heterogeneous. Six major cell clusters with different chromatin profiles suggested that the same type of perturbation induced a heterogeneous but overlapping range of chromatin responses. The proportion of the cells giving one or the other of the six responses varied between the conditions. To investigate the potential biological consequences of these differences, we extracted the list of differentially accessible peaks for each cluster. We assigned the closest genes to each peak and performed a gene ontology analysis on the gene lists. We focused on the first six clusters, because they contained most of the cells for all conditions. Results (Fig S5) showed that all clusters displayed high accessibility of genomic regions associated with the biological process typical for cells in growth and division, such as "positive regulation of cellular biogenesis," "regulation of cellular component size," "cell adhesion," or "cell communication." In clusters n°3 and 5, lymphoid- or myeloid-associated processes were also detected ("T-cell differentiation," $P_{adj}$ = 2.64 × $10^{-23}$; "lymphocyte differentiation," $P_{adj}$ = 2.64 × $10^{-23}$; "mononuclear cell differentiation," $P_{adj}$ = 4.80 × $10^{-22}$; "myeloid leukocyte activation," $P_{adj}$ = 5.91 × $10^{-14}$). It appears therefore that the cells' response to the metabolic perturbations is partially overlapping and heterogeneous with a varying fraction of cells providing a similar response in each condition. Here again, the response to AOA appears substantially different than the other inhibitors. It is worth reminding that, at this stage, despite the differences between conditions, the number of accessible promoters in the cells exceeded largely the maximal number of genes expressed in individual cells. The promoter accessibility is not the limiting factor responsible for restriction of gene transcription. However, the different chromatin profiles induced by the metabolic perturbations may channelize the subsequent evolution of the chromatin to later stages when the accessibility becomes a restrictive factor for gene transcription.

To verify whether this is indeed the case, we analyzed the transcriptomes by single-cell mRNA sequencing of the cells combined with epitope detection (CITE-seq) at 96 h post-stimulation. At this time point, the cells in each condition had recovered from the initial perturbation, and entered a phase of growth and proliferation (Fig 1C), and some of them went through the first steps of fate commitment (9). We analyzed 5,000 cells from each condition using the Chromium 10X technology. After quality control filtering, normalization, and alignment of the sequences on the genome, the data from the control and the perturbed conditions were integrated, and we visualized the structure of the cell populations using the usual dimension reduction approach (UMAP) (Fig 3A). The gene expression profiles generated in response to AOA, 2-DG, and DON were heterogeneous but partially overlapped with the control and each other. Overall, 17 clusters with varying numbers of cells were identified (Fig 3B). The distribution of the control and treated cells in the clusters was uneven (Fig 3C). Four clusters were entirely composed of AOA-treated cells (clusters n°8, 11, 14, and 15). 10 of the 17 clusters contained no AOA-treated cells, and two had only a small contribution of them. 2-DG–treated cells were overrepresented in clusters 2 and 10 but almost absent from clusters 0, 6, and 9. DON-treated cells were overrepresented in clusters 1 and 5, underrepresented in cluster 2, and almost similar to control cell distribution in other clusters. Each cluster had its own set of differentially expressed genes (Fig S6). Therefore, the cell's chromatin response to the metabolic perturbations at early stages was passed on to the gene expression profile at 96 h and generated a heterogeneous transcriptomic response dependent on the type of the metabolic stress. The mRNA profile of the AOA-treated cells showed the highest difference compared with the other conditions.

To assess the abundance of the CD34 and CD133 proteins on the cell's surface simultaneously with the RNA sequencing, we used CITE-seq (Fig S7A). Remarkably, the expression of these markers was the highest in clusters 0, 6, and 9 where AOA- and 2-DG–treated cells were absent or underrepresented (Fig S7B). These clusters, essentially composed of control and DON-treated cells, may represent the cells with phenotypes close to the pluripotent progenitor cells.

To better characterize the differences between the conditions, we performed pathway and GO analyses of the transcriptomes. We assessed pathway-level differences between the control and metabolically perturbed cells using ReactomeGSA (20). This approach performs the differential expression analysis directly on the pathway level. It detects small synchronous changes within a pathway that may reveal a biologically important effect. The results shown in Fig 3D indicated that AOA-treated cells displayed very different pathway activities compared with all the other conditions. Pathways related to "integration of energy metabolism," "cellular response to starvation," "glycogen breakdown," and "amino acid transport across the plasma membrane" were significantly upregulated in the AOA-treated cells. Pathways related to epigenetics such as "DNA methylation," "acetylation," or "pyruvate metabolism and citric acid cycle," "heme biosynthesis," "pentose phosphate pathway," among others, were down-regulated. These changes are indicative of a deep metabolic reorganization and difficulty in mobilizing energy resources. Interestingly, DON-treated cells presented a very different profile with, among others, upregulated "epigenetic regulation of gene expression," "DNA methylation," and "acetylation" pathways, and down-regulated "glutamate and glutamine metabolism." Without surprise, 2-DG–treated cells showed a different profile with an increase in

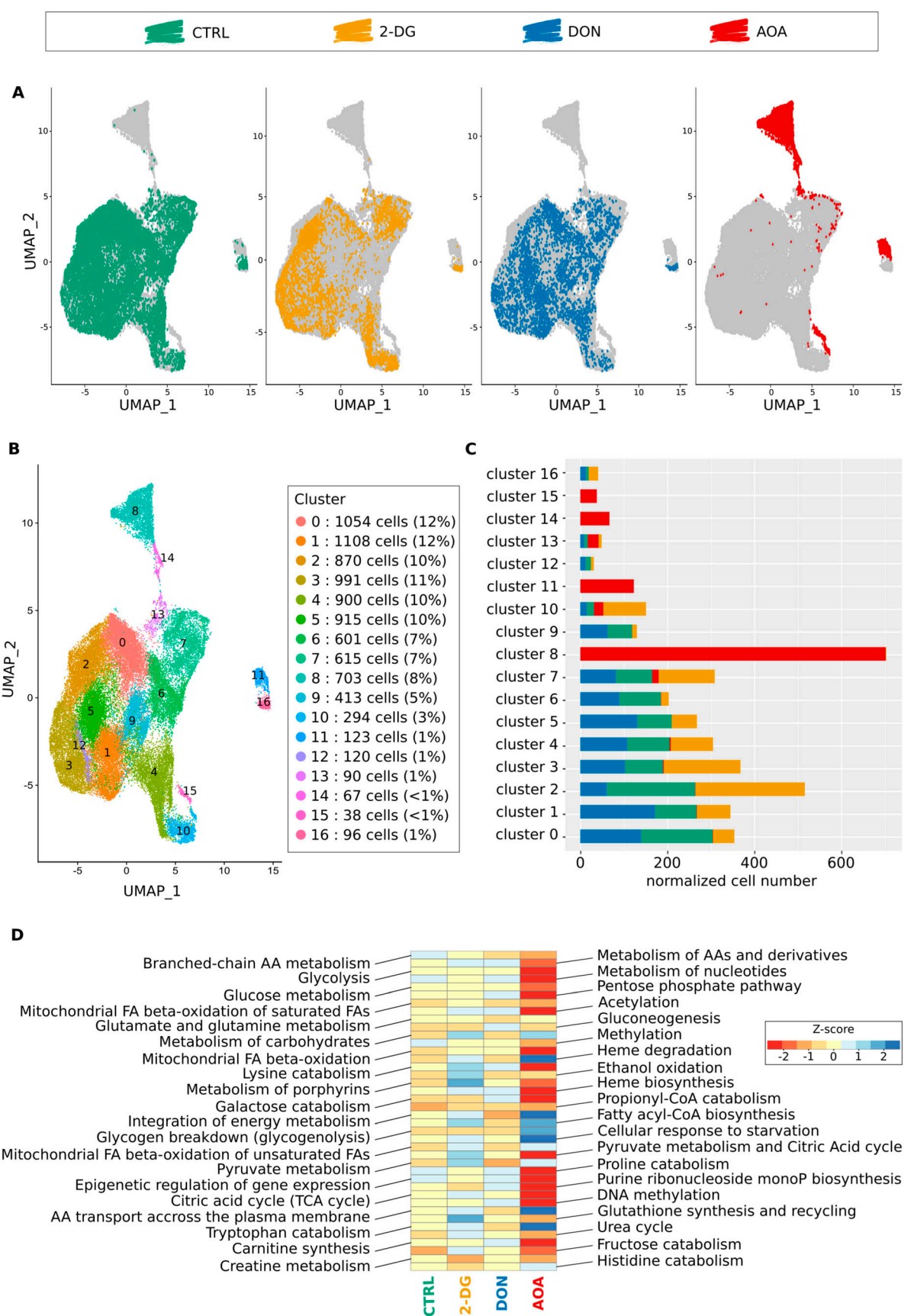

"glutathione biosynthesis," "metabolism of porphyrin," or "heme biosynthesis." Overall, the largest difference was observed between the AOA condition and the control. The GO analysis of differentially expressed markers between conditions (Table S3) showed that compared with control cells, 2-DG–treated cells displayed enriched gene expression in categories related to erythrocyte activity ("oxygen transport," $P_{adj}$ = 0.002; "hemoglobin complex," $P_{adj}$ = 3 × 10$^{-04}$; "oxygen carrier activity," $P_{adj}$ = 5 × 10$^{-04}$; "oxygen binding," $P_{adj}$ = 0.003; "heme binding," $P_{adj}$ = 0.03), whereas those related to monocytes or T lymphocytes were underrepresented ("regulation of monocyte differentiation," $P_{adj}$ = 0.008; "monocyte differentiation," $P_{adj}$ = 0.02; "T-cell differentiation," $P_{adj}$ = 0.02). In DON-treated cells' transcriptome, LT cell–associated terms seem numerous in the underrepresented categories ("regulation of activated T-cell proliferation," $P_{adj}$ = 0.03; "activated T-cell proliferation," $P_{adj}$ = 0.03; "T cell–mediated immunity," $P_{adj}$ = 0.04; "lymphoid progenitor cell differentiation," $P_{adj}$ = 0.04). However, a wide variety of terms associated with other lineages can be found in the overrepresented list, such as markers related to oxygen transport ("oxygen carrier activity," $P_{adj}$ = 0.008; "oxygen binding," $P_{adj}$ = 0.01; "oxygen transport," $P_{adj}$ = 0.004), megakaryocyte and platelet differentiation ("megakaryocyte differentiation," $P_{adj}$ = 0.01; "platelet aggregation," $P_{adj}$ = 0.01), leukocyte activities ("positive regulation of leukocyte cell–cell adhesion," $P_{adj}$ = 0.04), and overall early differentiation ("hematopoietic stem cell differentiation," $P_{adj}$ = 0.02). Markers for oxygen transport and megakaryocyte differentiation were underexpressed in AOA-treated cells ("oxygen transport," $P_{adj}$ = 0.009; "hemoglobin complex," $P_{adj}$ = 1 × 10$^{-04}$; "regulation of megakaryocyte differentiation," $P_{adj}$ = 0.02), whereas those related to leukocyte cells were largely enriched with around 20 associated terms extracted ("myeloid leukocyte migration," $P_{adj}$ = 0.02; "regulation of leukocyte cell–cell adhesion," $P_{adj}$ = 0.04; "leukocyte-mediated immunity," $P_{adj}$ = 6.4 × 10$^{-08}$). The enriched categories remain quite diverse, with not only references to the immune system ("antigen processing and presentation," $P_{adj}$ = 1 × 10$^{-06}$) but also references to LT cells both in enriched and in non-enriched categories, suggesting heterogeneity within the population. These observations showed that the cells adopted very different and complex gene expression strategies to overcome the constraints and start to proliferate.

Collectively, the ATAC-seq and RNA-seq studies revealed that early chromatin response to metabolic perturbation is followed by a reinforced transcription response at a later stage. Even though the chromatin opening is widespread in all conditions at 24 h, we observed certain heterogeneity between and within the conditions. More particularly, the ATAC profile in AOA-treated cells was markedly different compared with the other conditions. This tendency was reinforced when the transcriptome of the four conditions was compared at 96 h. To evaluate the potential long-term consequences of this metabolic priming, cytometry experiments were carried out at later time points.

## Transitory metabolic perturbation has long-term effects on cell differentiation

After the initial period of 96 h in the presence of inhibitors, cells were transferred to a fresh inhibitor-free culture medium for an additional 10 d. To assess the lineage commitment of the cells, the expression of several hematopoietic lineage markers (CD133, CD14, CD15, CD19, CD36, CD41, CD45) was measured by flow cytometry on days 4, 7, 10, and 14. The analysis on day 4 confirmed the CD133 expression pattern detected by cytometry (Fig 1B) and the high mortality rate in AOA- and 2-DG–treated cells already observed by time-lapse microscopy (Fig 1C). We found that the higher-than-normal mortality rates also persisted after day 4, even if the inhibitor was no longer present in the culture medium (Fig 4A). Importantly, we also discovered that the specific nature of transient metabolic perturbations preceding the initial fate determination influenced the differentiation pathways that the cells followed after the inhibitors were removed.

The dynamic evolution of individual markers and their combinations is shown in Figs 4 and 5. In addition to the high mortality all along the experiment (Fig 4A), AOA-treated cells showed the biggest differences compared with the controls (Fig 4B). The CD133 level was already lower than in the controls and other treated cells on day 4 and remained low on day 7. AOA-treated cells exhibited relatively low expression levels of CD45 and CD36, and a fluctuating dynamic of CD15. The proportion of cells with the combination of CD45$^+$ and CD36$^-$ associated with leukocyte (granulocyte and lymphocyte) differentiation was lower than in the control on days 4 and 7, but increased later. Within this group, the CD45$^+$CD36$^-$CD15$^+$ cells typical for the granulocyte path were overrepresented during the first two time points and approached the control proportion in the later stages. The fraction of cells with typical monocytic markers was also variable. The CD45$^+$CD36$^+$ cells consistently showed lower representation at all time points compared with the control condition. However, the proportion of CD14$^+$ monocytes within this population was significantly elevated on days 4 and 7 but dropped below the control level on days 10 and 14. Such a variable expression may reflect not only a change in the marker expression in the cells but also a variable proliferation capacity of the subpopulation. No erythrocyte progenitors were detected in AOA-treated cells; however, a substantial proportion of CD45$^-$CD36$^-$ cells appear to persist over time. Overall, the differentiation of the AOA-treated cells appears to be truly disturbed even after the removal of the metabolic inhibitor.

**Figure 3.  Differential gene expression in control and inhibitor-treated cells at 96 h as determined by scRNA-seq transcriptome analysis.**
**(A)** UMAP of the merged scRNA-seq data. Each dot represents a single cell. The same UMAP is reproduced four times to highlight the cells of only one of the four conditions at the same time. The color code is indicated at the top of the figure. All the other cells are gray. **(B)** Same UMAP map as in (A) is represented to allow the identification of individual clusters. The clusters were numbered from the largest to the smallest. The numbers, the color codes, and the relative size of the clusters are indicated in the inset on the right. **(C)** Histogram of the cluster compositions. Color codes for the control and inhibitor-treated conditions are identical as in (A). Note that although the cell populations treated with DON and 2-DG overlap with the control population, AOA-treated cells appear to have developed entirely distinct expression profiles. **(D)** ReactomeGSA's pathway analysis of 40 pathways of interest. The pathway names are given on the two sides of the heatmap, and the color code for the Z-score of the expression levels is on the right inset. Note the very different pathway profile of the AOA-treated cells.

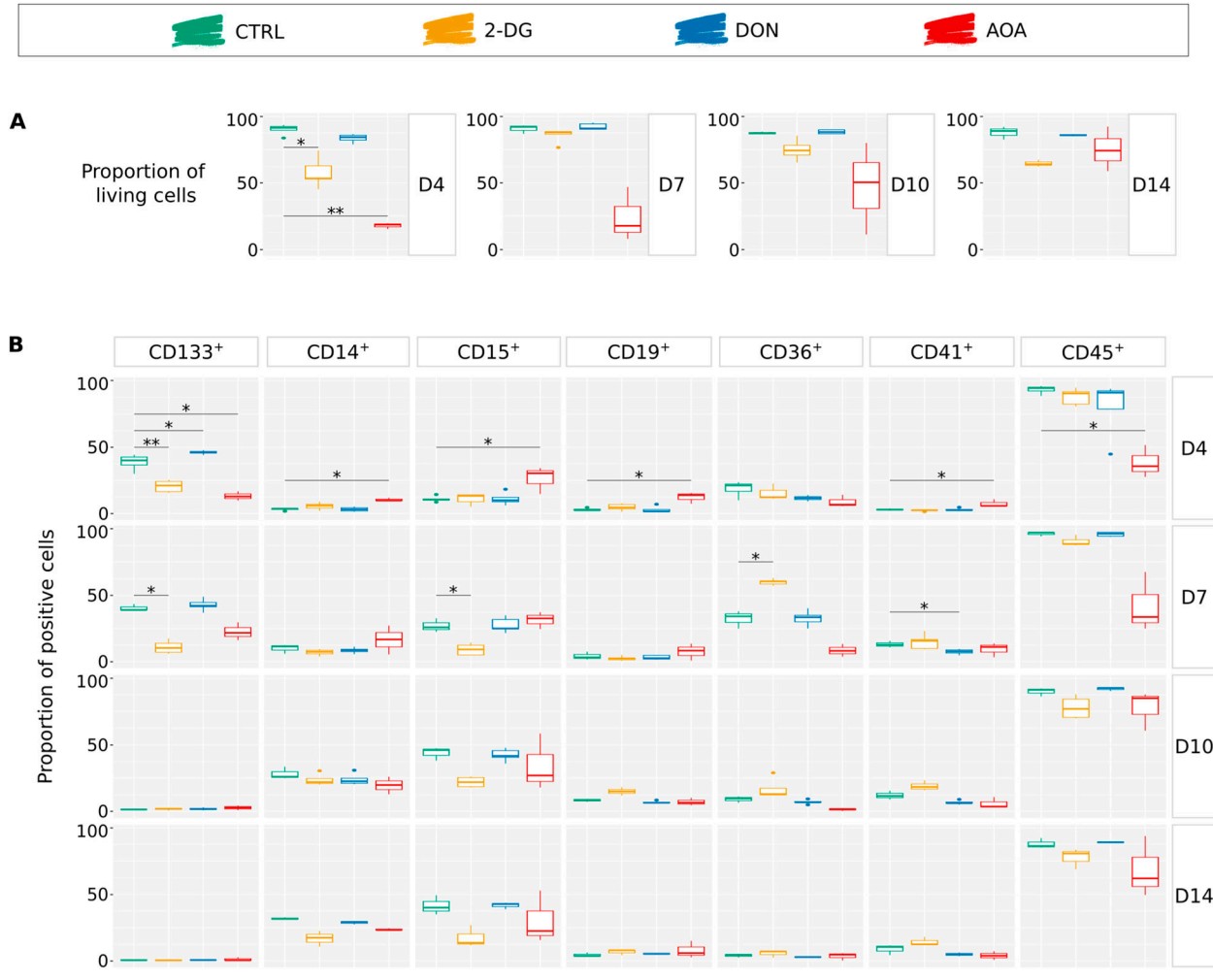

**Figure 4.  Analysis of the cell differentiation potential in long-term in vitro cultures at days 4, 7, 10, and 14—simple markers.**
Results of several experiments are summarized as boxplots, displaying the proportion of positive cells for each individual marker. The Wilcoxon statistical test was performed on pairwise comparisons (control versus inhibitor). **(A)** Proportion of living cells for each condition over time. AOA-treated cells displayed high heterogeneity of sensibility in response to the inhibitor compared with the other conditions. **(B)** Proportion of cells displaying the individual markers on their membrane. CD133 is considered as a multipotent marker. CD14 and CD15 are associated with myeloid cells, respectively monocytes/macrophages and granulocytes/monocytes. CD19 is a marker commonly found on B lymphocytes and CD36 mostly on erythrocyte progenitors. CD41 is characteristic of platelets, and CD45 is present on the surface of all nucleated hematopoietic cells.

Transient 2-DG treatment also strongly influenced the differentiation potential of the cells. These cells also had a higher death rate than the control cells. The CD36 marker showed a sharp transient increase on day 7 with more than 50% of the cell population expressing it. It is worth mentioning that CD36 is required for free fatty acid uptake and transition from glycolysis toward β-oxidation of fatty acids by undifferentiated hematopoietic cells (21). The expression of CD36 may represent a metabolic adaptation strategy that helps avoiding energy shortage because of the inhibition of glycolysis by increasing fatty acid oxidation (22). CD41$^+$, CD45$^-$CD36$^+$, and CD45$^-$CD36$^-$ cells are slightly more represented compared with controls. The most significant effect of the metabolic perturbation was observed on marker combinations representative of the myeloid and the lymphoid branches. There is a significant decrease in the number of CD45$^+$CD36$^-$ cells starting from day 7 compared with the control group, although this decrease becomes less pronounced over time. However, the CD45$^+$CD36$^-$CD15$^+$ granulocytes remain underrepresented within this population until day 14. In addition, there is a notable increase in the number of CD45$^+$CD36$^+$ cells on day 7. However, unlike what was observed with AOA treatment, the CD14$^+$ cells within this population are underrepresented.

As we have shown earlier, DON-treated cells recovered normal morphological phenotype and proliferation capacity from the initial metabolic stress caused by the inhibition of the glutaminase around 96 h (Fig 1B and C). Although we observed minor effects on the chromatin structure at 24 h (Fig 2) and on the gene expression profile at 96 h (Fig 3), these cells showed an overall similar pattern of commitment as the control cells as witnessed by the cell surface marker distributions (Figs 4 and 5). A potential explanation for this full recovery could be the fact that glutamine is a non-essential amino acid.

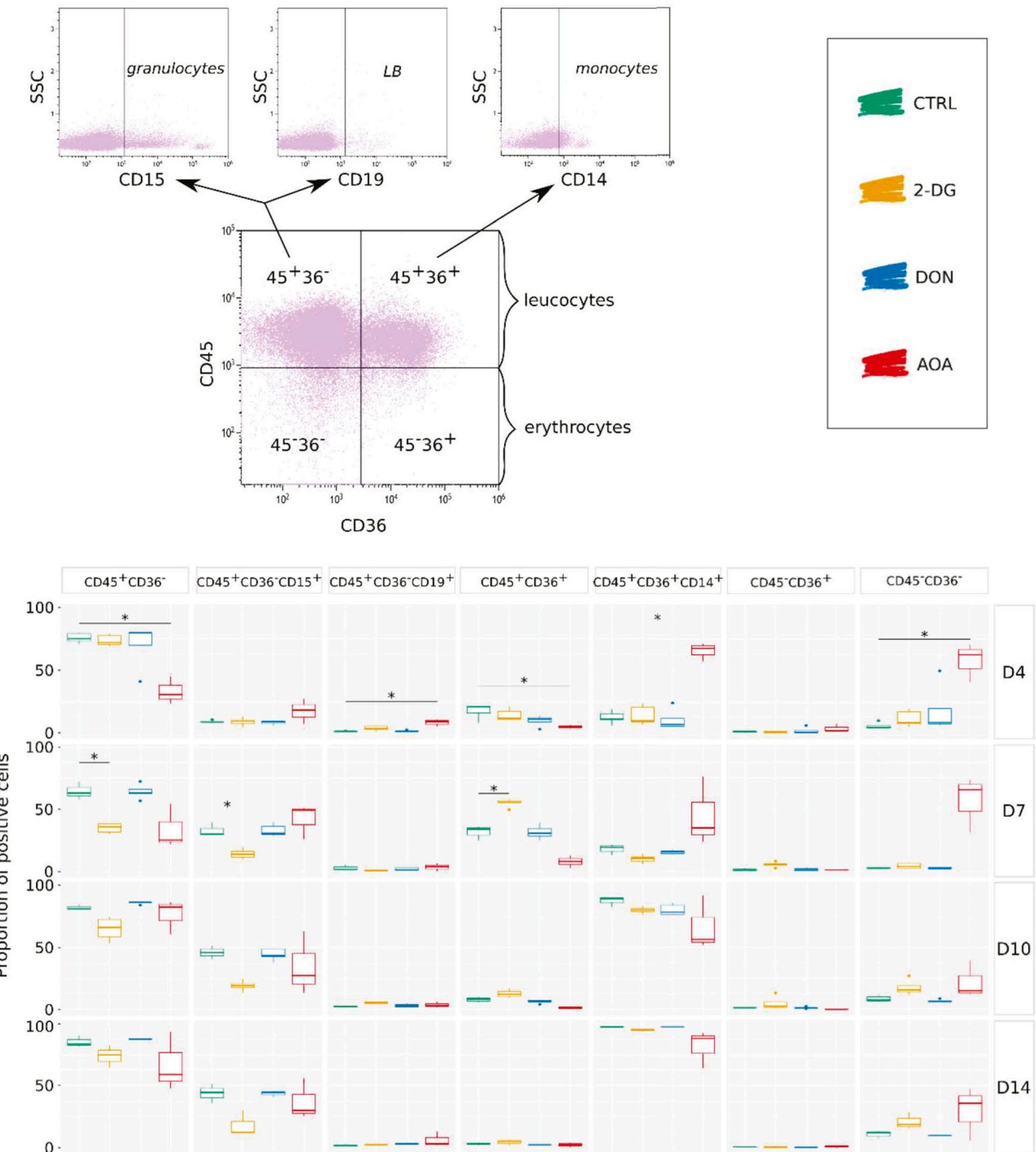

**Figure 5. Analysis of the cell differentiation potential in long-term in vitro cultures at days 4, 7, 10, and 14—combined markers.**
Results of several experiments are summarized as boxplots displaying the proportion of positive cells for each combination of markers: CD45⁺CD36⁻ for granulocytes and lymphocytes, CD45⁺CD36⁻CD15⁺ for granulocytic cells, CD45⁺CD36⁻CD19⁺ for B lymphocytes, CD45⁺CD36⁺ for monocytes and macrophages, CD45⁺CD36⁺CD14⁺ for monocytes, CD45⁻CD36⁺ mostly for erythrocyte progenitors, and CD45⁻CD36⁻ for mature erythrocytes. The Wilcoxon statistical test was performed on pairwise comparisons (control versus inhibitor).

These observations are in line with the pathway and GO analyses of the transcriptomes shown earlier. They indicate that a transitory perturbation of important metabolic pathways of glucose and glutamine use can modify the cell's capacity to divide and differentiate long after the initial trigger is removed.

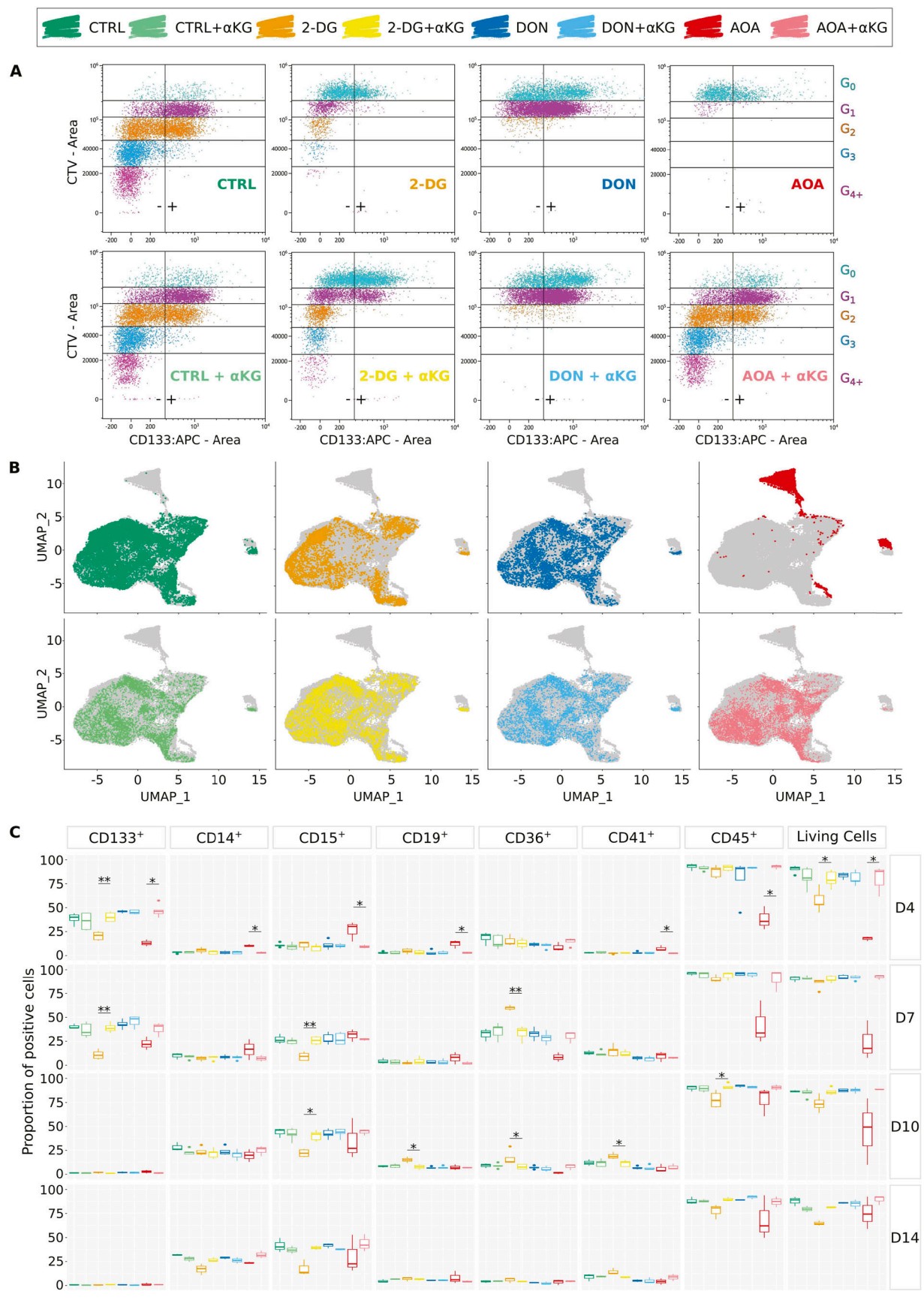

## α-Ketoglutarate can compensate for some but not every metabolic perturbation

Overall, general cellular responses to the three tested metabolic perturbations were highly distinct. Throughout the initial 96-h period of cell culture, it became apparent that cells encountered greater challenges in overcoming the perturbations caused by the inhibition of glutaminase with DON or transaminases with AOA compared with the perturbation affecting glucose metabolism induced by 2-DG (Fig 1B and C). However, beyond this phase, DON-treated cells appeared to recover and acclimate to their new environment. We observed no substantial variations in terms of fate commitment compared with control cells over the next 10 d of in vitro culture. In contrast, 2-DG– and AOA-treated cells remained durably affected.

Glucose and glutamine are metabolized through two separate pathways. 2-DG inhibits the first pathway, whereas DON and AOA impede the latter one. Both pathways feed in the TCA cycle and generate α-ketoglutarate. This metabolite can be either oxidized and contributed to the ATP production or metabolized reductively to generate acetyl-CoA. In addition, it plays a crucial role as a bridge connecting energy metabolism and chromatin-modifying epigenetic mechanisms, serving as a substrate to DNA- and histone-demethylating enzymes. To highlight the role of this metabolite, we conducted rescue experiments with αKG to ascertain whether the inhibitors' effects could be mitigated or compensated for. To do this, we supplemented the culture medium with αKG and repeated the experiments conducted at 96 h and beyond.

When analyzed by cytometry at 96 h, the CD133 and CD34 distribution and proliferation capacity of the αKG-supplemented AOA-treated cells were essentially indistinguishable from the normal control cells (Fig 6A and Table S1). The study of transcription profiles by scRNA-seq showed a remarkable restoration in gene expression profiles in AOA-treated cells, marked by the disappearance of the specific AOA profiles and a return of most of the profiles seen in the control condition. However, the transcription profile of the cells remained partially different. αKG-supplemented AOA-treated cells are nearly absent from cluster n°2 and over-represented in cluster 4 (Fig 6B). The metabolic pathway analysis of the scRNA-seq data also indicated a strong αKG effect. The activity of the gene pathways related to various metabolic processes down-regulated in the AOA-treated cells raised to a higher level than in the control cells with αKG (Fig S8). The GO analysis of the differentially expressed genes also confirmed that the effect of αKG is more complex than simple compensation of the transaminase inhibition. Markers for oxygen transport that were underexpressed in AOA-treated cells became now positive markers of AOA+αKG cells

("oxygen transport," $P_{adj}$ = 0.001; "hemoglobin complex," $P_{adj}$ = 4 × $10^{-04}$; "oxygen binding," $P_{adj}$ = 0.005). Conversely, several enriched gene categories in AOA-treated cells related to lymphocyte and leukocyte differentiation became now part of the negative markers ("T-cell activation," $P_{adj}$ = 4 × $10^{-06}$; "regulation of leukocyte cell–cell adhesion," $P_{adj}$ = 5 × $10^{-07}$). Some categories remained positive in both conditions ("positive regulation of lymphocyte activation," $P_{adj}$ = 0.03; "dendritic cell migration," $P_{adj}$ = 0.007; "myeloid cell homeostasis," $P_{adj}$ = 0.03). Contrary to the transcription profile, αKG completely erased the effects of AOA treatment on the long-term differentiation capacity of the cells. We observed similar cell surface marker levels and combinations as on the control untreated cells (Figs 6C and S9). The fact that αKG could rescue the proliferation and differentiation capacity of AOA-inhibited cells without fully restoring the gene expression profile identical to the untreated cells shows that alternative gene expression profiles and pathways can lead to a similar biological outcome and suggests a complex relationship between the metabolism, gene expression, and differentiation.

αKG improved the proliferation capacity of the 2-DG–treated cells and restored the expression pattern of CD133 (Fig 6A). As shown by the UMAP representation, the gene expression pattern of the 2-DG+αKG cells was also similar to the control cells. The introduction of αKG allowed the restoration of cell populations that were otherwise diminished when exposed solely to the inhibitor (clusters 0, 6, and 9), and reinstated a more balanced distribution of cells within clusters. The pathway and GO analyses showed that the activity of several pathways was reversed by αKG (Fig S8 and Table S4). Several gene ontology categories linked to erythrocyte activities, which exhibited high activity in 2-DG–treated cells, were down-regulated in 2-DG+αKG cells ("oxygen transport," $P_{adj}$ = 1.12 × $10^{-06}$; "oxygen carrier activity," $P_{adj}$ = 7.8 × $10^{-08}$; "oxygen binding," $P_{adj}$ = 1.3 × $10^{-06}$; "erythrocyte differentiation," $P_{adj}$ = 0.002). Markers associated with other hematopoietic functions were enriched ("B-cell activation," $P_{adj}$ = 0.01; "leukocyte-mediated immunity," $P_{adj}$ = 0.02; "macrophage activation," $P_{adj}$ = 0.03; "platelet activation," $P_{adj}$ = 0.04). The in vitro differentiation test also demonstrated that αKG reversed the effect of 2-DG on the differentiation capacity of the cells. The evolution of cell populations with different cell surface marker combinations was similar to the control cells (Fig S9).

Contrary to AOA and 2-DG, αKG supplementation had no effect on the proliferation and cell surface marker distribution of the DON-treated cells during the first 96 h (Fig 6A). αKG had only a slight effect on the pathway expression pattern of the DON-treated cells (Fig S8). As indicated earlier, DON-treated cells returned to normal proliferation and differentiation after the lag period. DON+αKG cells followed the same pattern (Figs 6C and S9).

**Figure 6.   Rescue experiments with αKG.**
**(A)** CD133 expression and generation tracking measured by flow cytometry at 96 h. Generations are indicated by the different colors. Positive and negative populations are delimited by a vertical line and –/+ signs. Detailed proportions of positive populations and MFI per generation for not only CD133 but also CD34 markers are available in Table S1. In addition to restoring viability, the addition of αKG in the culture medium seems to completely counteract the inhibition induced by AOA. **(B)** UMAP of the scRNA-seq data with or without αKG. Each dot represents a cell. The gray profile represents the shape of the total population when all conditions are merged. Cells treated with different inhibitors are highlighted by their corresponding colors. Individual clusters are not highlighted; please refer to Fig 3B. The addition of αKG restored the gene expression profile of 2-DG–treated cells, but only partially in AOA-treated cells. **(C)** Proportion of positive cells for individual markers from the long-term cytometry experiment at days 4, 7, 10, and 14. The Wilcoxon statistical test was performed on pairwise comparisons between condition and condition + αKG. All markers seem to be expressed at the same level as in control when αKG is added with the inhibitors.

Finally, we examined the effect of αKG supplementation alone on the control cells. We observed no substantial effect neither on the cell proliferation or differentiation. Nevertheless, transcriptome analysis showed a slight effect on pathway activity (Fig S8).

In summary, αKG was able to compensate for the short- and long-term effects of AOA and 2-DG. The rescue of the AOA-treated cells was the most spectacular. The partial inhibition of the transaminases strongly impacted all aspects of cellular physiology (proliferation, differentiation). These effects were fully reversed, but without fully reverting AOA's effect on gene expression. Less contrasted, but similar, tendency was observed with 2-DG. Although αKG had no detectable effect on the control cells, it did influence their gene expression pattern also. The observations support the idea that not only energy metabolism but also many other essential physiological processes rely on αKG.

# Discussion

Taken together, our data find that early, partial, and transitory inhibition of enzymes of the glucose or glutamine pathways generates perturbations of core cellular metabolism. These early perturbations trigger changes that interfere with the process of global chromatin opening typically occurring during the first 24 h after the cells are stimulated. This interference is dependent on the type of metabolic stress. The gene expression profiles of cells are altered and show clear signs that the process of cell fate choice is affected. The long-term culture of the cells confirms that their differentiation capacity has been permanently modified by the early-stage transitory metabolic stress. Importantly, the response given by the cells to the transitory perturbation is heterogeneous at the level of the chromatin, gene expression, and differentiation capacity.

A crucial question is whether the observed effects from inhibitors are linked to modifications in the cell fate choice process or result from the selective survival and amplification of a pre-existing minor cell population. The CD34[+] cell population is inherently heterogeneous, and we used inhibitor concentrations close to the IC50 values. Therefore, the surviving cells exhibited greater resistance, likely because of their more resistant metabolic setup. Metabolic MS analysis was conducted on a bulk population, preventing conclusions at the single-cell level. However, time-lapse, cytometry, single-cell ATAC, and RNA-seq analyses all provide information at the single-cell level. ATAC-seq revealed initial differences between control and treated cells ~12 h after stimulation. By 24 h, 16 different subsets of cells were identified using single-cell ATAC-seq chromatin accessibility profiling. All four conditions were represented in all subsets in variable proportions. Previous studies (1, 9) indicated that at 24 h, these cells could not be clustered into distinct groups based on their gene expression patterns, suggesting that chromatin changes precede gene expression changes by several hours. Notably, at the time of analysis, these cells had not undergone division yet. Time-lapse microscopy revealed that the first division occurred in control and 2-DG cells 24 h later, whereas in DON and AOA cells, it occurred only around 72 h later. At this point, single-cell RNA-seq data clustering identified 17 different

subsets of cells. Particularly, AOA cells exhibited a distinctly different gene expression pattern, forming separate clusters. Based on these observations, we think that although some selection occurs during the initial hours, the differences observed between the inhibitors cannot be solely explained by it. Instead, chromatin differences between cells appear before the first division of the cells surviving the initial shock. These differences then gradually evolve over the initial 96 h. The inhibitors were removed at this point, and the cells primed by the different inhibitors were subsequently cultured under identical conditions. It is likely that cells exhibiting differential gene expression patterns possessed varying proliferation capacities, contributing to the observed evolution of cell populations as detected on days 7, 10, and 14.

Several important points arise from the observations. Restricting glucose access through partial inhibition of glycolysis by 2-DG reduces the capacity of the cells to survive and proliferate. However, presumably thanks to the availability of carbon and energy sources provided by glutamine, the cells can rescue cellular functions. Glutamine is not only a carbon source, but it is also the major source of nitrogen for nucleotide and amino acid biosynthesis. Because glucose does not contain nitrogen, it cannot replace glutamine. Given that glutamine is non-essential, cells could potentially adapt to decreased external availability by redirecting their metabolism toward the use of alternative amino acids. The time required for the metabolic adaptation and recovery of the biosynthetic capacity from altered resources may explain the long lag before the first division of the DON- and AOA-treated cells. Nevertheless, there is a notable difference between the effects of DON and AOA. The latter impacted stronger the physiology of the surviving cells, and they never recover completely their capacity to grow and divide. Inhibition of the transaminases in the CD34[+] cells appears therefore more difficult to compensate for by alternative strategies than the inhibition of the upstream step of the glutamine use. Indeed, transaminases catalyze a key step that links the glutamine metabolism to the central energy metabolism through the deamination of glutamate resulting in αKG. AOA also inhibits the malate–aspartate shuttle and blocks the electron flow between the cytoplasm and the mitochondria resulting in reduced ATP synthesis. Aspartate was indeed shown to be essential for hematopoietic cell differentiation and entirely dependent on cell-autonomous synthesis (23). Importantly, external αKG could fully compensate for the effect of transaminase inhibition by AOA, showing that the effect of AOA is essentially conveyed by this intermediate of the TCA cycle. αKG is well known for its role as a substrate of DNA- and histone-demethylating enzymes (24, 25). Its involvement in epigenetic mechanisms is likely to be the key to its role in stem cell function and cell differentiation (26, 27, 28, 29). The role of αKG in the self-renewal and differentiation of hematopoietic stem cells is also widely recognized (30). The disparity in chromatin impact between AOA and DON, along with AOA's responsiveness to αKG, might reside from the configuration of the glutamine metabolic pathway. DON's inhibition effects on αKG production can be more readily offset by other amino acids, given that glutamine is a non-essential amino acid. This aligns with the observed recovery of surviving cells after an initial setback, where they subsequently resume their proliferation and differentiation after a brief lag period. Conversely, compensating for the inhibition caused by AOA

is more challenging because of the direct involvement of transaminases in $\alpha$KG production.

Cell differentiation is typically explained as a process driven by the dynamics of gene regulatory networks (31, 32, 33, 34). This description is undoubtedly incomplete because it dismisses the required bioenergetics component of the process. It is separately understood that the cell's survival is intricately linked to the continuous flow of energy. Yet, this is rarely included in the explanatory scheme, nor has this been considered as a driver of cell state/fate changes. The energy flow is ensured by the stepwise oxidation of the carbon source molecules taken up from the environment with the concomitant conversion of the chemical energy into small high-energy metabolites ready to be used in biochemical reactions. Because the availability of the carbon sources is out of the cell's control, their response to a decreased energy supply is limited to switching to a new energy source. This transition corresponds to a differentiation process. Because metabolic adaptation is the absolute requirement for the survival of every cell, the insufficiency of the bioenergetics resources alone would represent a universal stimulation for differentiation. Although the biochemical links between the nutrient availability, key metabolites, epigenetic mechanisms, and cell differentiation are now well recognized, the exact character of this connection remains unclear. As recently formulated, "The observation that TCA cycle metabolites can control $\alpha$KG-dependent dioxygenases and cell fate raises major questions, including how a nonspecific signal such as TCA cycle intermediate abundance can lead to a specific outcome in cell fate" (35). From our data in this study, we infer that the *nonspecific* metabolic stress acts as a primary "activating signal" required to generate a variety of non-specific, dynamically fluctuating chromatin and gene expression responses. This can represent the multilineage primed state in hematopoietic cells (36, 37, 38). Indeed, similar transcription responses have been observed in all the other cell differentiation models examined (39). This exploratory dynamic generates variants with distinct gene expression patterns that can restore the energy flux by sampling any alternative metabolic sources available in the environment. Different cells can reach different "solutions," that is, different phenotypic states in the same environment, and form a cellular ecosystem that supports metabolic cooperation but not necessarily free of competition. Such cooperative cellular setups that characterize the multicellular organisms are very well known in the unicellular world (40, 41, 42, 43). The specific outcome of the non-specific stress response also relies on intracellular factors commonly referred to as cellular or epigenetic memory. The degree of chromatin opening in response to metabolic stress also depends on the nature and density of pre-existing repressive modifications reflecting the life history of the cell. It is worth mentioning that, although not considered in this study, post-translational modifications of many cellular proteins unrelated to chromatin but dependent on the same sentinel metabolite substrates also could potentially contribute to the cellular memory process.

It is becoming increasingly popular to consider cell differentiation as a process of selective stabilization of gene expression profiles generated by spontaneous stochastic variation of gene transcription analogous to the process of evolution (44, 45, 46, 47, 48). Others compared differentiation with a "learning process" (49, 50). The initial theory has been further developed by including energy metabolism as an essential part of the process (51, 52). According to possible scenarios based on these considerations, metabolic stress induces a non-specific and highly variable response. Extrinsic and intrinsic constraints and contingencies selectively stabilize those responses that allow better survival. The epigenetic mechanisms stabilize and ensure the transmission of the cellular state until the next metabolic stress. Although speculative, the observations presented in this study provide arguments in favor of this theoretical model and integrate our previous knowledge of the underlying molecular mechanisms in a coherent way. They demonstrate, using a widely used experimental model, that the significant metabolic stress may be the initial event, immediately followed by global chromatin decompaction. The exact nature of the metabolic stress and the cells' intrinsic properties together canalize the chromatin modifications that generate differences in gene expression at a later stage. This is sufficient to impact the subsequent cell fate trajectories even if the initial metabolic stress has ceased.

There are many observations making causal links between metabolism and cell differentiation or stem cell properties in the hematopoietic and other cellular systems (16, 18, 29, 53). Typical explanations for the above link each observed effect to very specific molecular mechanisms acting in specific circumstances (54, 55). Our model provides a more generalizable explanation for all cases. In addition, we can also speculate on an evolutionary perspective for cell differentiation. In fact, multicellularity and cell differentiation appeared multiple times during evolution (56, 57), before specific molecular mechanisms emerged, suggesting that their role in the process of cell differentiation could be supportive. A popular view of cell differentiation is to consider the stable cell phenotype as an attractor in the multidimensional Waddington-type epigenetic landscape defined by the gene regulatory network (32, 33, 34, 58). Differentiation is seen as a transition between two attractors. Our model integrates this vision where the attractors are defined, in addition to the gene regulatory network, by the metabolic fluxes and the influence they have on the dynamics of the epigenetic chromatin states. As a result, the Waddington landscape is not simply determined by the intrinsic properties of the gene network, but becomes a dynamic process, resulting from the interactions between intrinsic and extrinsic constraints.

# Materials and Methods

### Ethics statement

Human umbilical cord blood was collected from placentas and/or umbilical cords from anonymous donors obtained from AP-HP, Hôpital Saint-Louis, Unité de Thérapie cellulaire, CRB-Banque de Sang de Cordon, Paris, France (authorization number: AC-2016-2759), or from Centre Hospitalier Sud Francilien, Evry, France, in accordance with international ethical principles and French national law (bioethics law no 2011–814) under declaration No DC-201-1655 to the French Ministry of Research and Higher Studies.

## Cell culture

Human CD34[+] cells were isolated from the umbilical cord blood of anonymous healthy donors. After an erythrocyte depletion step using dextran from *Leuconostoc* spp. ($M_r$ 450000-650000; Sigma-Aldrich), cells were separated by density centrifugation using Ficoll (Biocoll; Merck Millipore). The mononuclear fraction was collected and enriched in CD34[+] by immunomagnetic beads using a manual MACS system (Miltenyi Biotec). After collection, enriched CD34[+] cells were frozen in a cryopreservation medium containing 90% of FBS (Eurobio) and 10% of dimethyl sulfoxide (Sigma-Aldrich) and stored in liquid nitrogen.

After thawing, cells were cultured at 37°C in a humidified 5% $CO_2$ incubator in a prestimulation medium supplemented with h-Flt3-L (50 ng/ml), h-IL-3 (10 ng/ml), h-SCF (25 ng/ml), and h-TPO (25 ng/ml) as previously described (1, 9). Metabolic inhibitors were added to the medium at their IC50 concentration determined experimentally. 6-Diazo-5-oxo-L-norleucine (DON) was used at a final concentration of 2.5 $\mu$M (ref. D2141; Sigma-Aldrich). 2-Deoxy-D-glucose (2-DG) was used at a final concentration of 1 mM (ref. D3179; Sigma-Aldrich). Amino-oxyacetic acid (AOA) was used at a final concentration of 1.9 mM (ref. C13408; Sigma-Aldrich). Some rescue experiments were performed with the addition of dimethyl $\alpha$-ketoglutarate in the inhibitor–medium mix at a final concentration of 5 mM (ref. 349631; Sigma-Aldrich). Inhibitors were stored according to the supplier's recommendations.

Metabolic inhibitors were added to the culture medium at time point 00 h of the experiments. For all time points less than 96 h of culture and for the time-lapse experiments, no medium change occurred. Nonetheless, the cytometry experiment on long-term cultures involved some experimental adjustments. After 96 h of culture with inhibitors as described above, the medium was changed, and all cells were cultured in the optimized prestimulation medium for long-term differentiation without inhibitors. Four more cytokines were added to the medium to allow unrestricted differentiation of the cells toward hematopoietic lineages: 1 U h-EPO, 10 ng/ml h-IL-6, 20 ng/ml h-GM-CSF, and 20 ng/ml h-G-CSF final concentration (PeproTech). Half of the medium was removed every 2 d, and the cell density was adjusted to 500,000 cells/ml with fresh medium (or 300,000 cells/ml before weekends).

## Time-lapse microscopy

The time-lapse microscopy protocol was previously described (9, 59, 60). Cells were cultured in a four-compartment dish (Ibidi Hi-Q4) combined with a PDMS microgrid array (Microsurfaces). Each compartment contained a different condition (control medium or metabolic inhibitors) allowing all the conditions to be assessed parallelly in the same experiment. Time-lapse acquisitions were performed with the inverted microscope Olympus IX83 combined with a thermostatic culture chamber from the CISA platform (CRSA). Images were acquired every 2 min using a 10X objective for 7 d. Microwells containing less than eight cells in the first image were considered in the analysis to allow cell tracking.

## Cytometry on short-term culture with generational tracking

After thawing, CD34[+] cells of three independent healthy donors were labeled separately using CellTrace Violet (CTV) Proliferation Kit (Invitrogen) to allow tracking of multiple generations with dye dilution. A sample of those parental generation cells were used to measure fluorescence intensity at time point 00 h. The other cells were incubated according to the conditions described earlier. At 96 h, cells were harvested and labeled using CD133-APC (clone AC133, dilution 1:50; Miltenyi Biotech) and CD34-PE (clone AC136, dilution 1:10; Miltenyi Biotech) cell surface antibodies after a non-specific sites' saturation step with Gamma Immune (dilution 1:2; Sigma-Aldrich). Isotype controls (Miltenyi Biotech), VersaComp compensation beads (Beckman Coulter), and negative controls were used for subsequent gating strategy. Viability was controlled using a 7-aminoactinomycin D marker (Invitrogen). The fluorescence intensity of CTV and antibodies was measured with the SP6800 cytometer (Sony) from the ImCy platform (Généthon).

## Cytometry on long-term cultures

Cytometry results represent at least three biological replicates for each time point on long-term culture and were obtained from a combination of seven experiments with mixes of 10 independent healthy donors. Cells were cultured in a prestimulation medium with or without metabolic drugs and αKG during the first 4 d and then in the optimized prestimulation medium without inhibitor or αKG as described before. A sample of cells were collected at days 4, 7, 10, and 14 to be analyzed by flow cytometry on CytoFLEX V5-B5-R3 Flow Cytometer (Beckman Coulter). After a saturation step with Gamma Immune (dilution 1:2; Sigma-Aldrich), cells were labeled with antibodies detecting the following cell surface markers: CD133-APC (clone 7, dilution 1:20; BioLegend), CD19-PE Cy7 (clone HIB19, dilution 1:80; BioLegend), CD36-PerCP-Cy5.5 (clone 5–271, dilution 1:40; BioLegend), CD45-BV421 (clone HI30, dilution 1:80; BioLegend), CD15-APC-Fire 750 (clone W6D3, dilution 1:80; BioLegend), CD14-BV785 (clone M5E2, dilution 1:80; BioLegend), CD41-BV605 (clone HIP8, dilution 1:40; BioLegend), and Zombie Aqua fixable viability dye (dilution 1:200; BioLegend). For each experiment, compensation beads (ref. 130-097-900; Miltenyi) were used for positive mono-labeling controls and unmarked cells for negative controls.

## Mass spectrometry

Cells of at least two independent healthy donors were mixed for each experiment and cultured during 24 h with or without the inhibitors, then collected for the extraction of metabolites. Cell suspensions were adjusted to 8 ml final volume and mixed with 42 ml of quenching solution (60% [vol/vol] methanol, 0.85% [wt/vol] ammonium bicarbonate, completed with Milli-Q water and pH adjusted to 7.4) cooled at −40°C. After centrifugation at 1,000*g* during 1 min and removal of the quenching solution by aspiration, cell pellets were resuspended in extraction buffer (80% ethanol) and vortexed vigorously to release all the intracellular metabolites. After another centrifugation and vortexing steps, supernatants were collected and centrifuged at 15,000*g* for 5 min. Supernatants were collected and dried using Speed Vacuum Concentrator 5301 (ref.530101197, 230 V, 1.4 A, 50 Hz, 310 W; Eppendorf) for 3 to 5 h. Metabolite extracts were kept at −80° before subsequent steps. Subsequently, targeted metabolites from distinct pathways (amino acids, tricarboxylic cycle, sugar phosphates) were measured either

with or without chemical derivatization, as described in detail earlier ([19]), using a QTRAP 6500 LC/MS/MS-based platform (AB Sciex).

## Bulk ATAC-seq

5,000 living cells of three independent healthy donors were sorted by FACS (Beckman Coulter) based on 7-AAD labeling. For library preparation, we used the Fast ATAC-seq protocol optimized for blood cells with minor modifications, as previously described ([1]). After quality control using Bioanalyzer (Agilent), libraries were sent to the CNRGH (CEA) for sequencing (NextSeq Illumina).

## 10X single-cell ATAC-seq

After 24 h of culture of a mixed cell suspension from six healthy donors, dead cells were removed using Dead Cell Removal Kit (Miltenyi Biotech). Samples were brought to the technical platform of the IMAGINE Institute (https://www.institutimagine.org/en), which performed the experiment and raw data processing. Nuclear isolation was carried out following the 10X Genomics protocol (CG000124), and nuclei were counted using Luna Automated Cell Counter (Logos). Libraries were prepared using Chromium Next GEM Single Cell ATAC Reagent Kit v1.1 (10X Genomics, protocol CG000209) from around 15,000 single nuclei for each condition. After the loading step on the microfluidic device, between 4,000 and 6,000 nuclei per sample were successfully encapsulated for further analysis at the single-cell level. After a quality control checkup with Qubit Fluorometer, libraries were then sequenced on Flowcell S1 on NovaSeq 6000 (Illumina).

## 10X single-cell RNA-seq—CITE-seq

Pools of cells from seven independent donors were cultured during 96 h with or without inhibitors and $\alpha$KG as described above. Cells were harvested, and dead cells were removed using Dead Cell Removal Kit (Miltenyi Biotech). Single-cell RNA-seq with feature barcoding technology for cell surface protein (CITE-seq) was then carried out following the Chromium Next GEM Single Cell 3′ protocol for v3.1 reagents from 10X Genomics (CG000206 Rev D) and TotalSeq-A Antibodies and Cell Hashing protocol from BioLegend (available online at https://www.biolegend.com/en-gb/protocols/totalseq-a-antibodies-and-cell-hashing-with-10x-single-cell-3-reagent-kit-v3-3-1-protocol). Cell hashing with hashtag oligos allows to mix different samples together after staining and, though, to reduce the number of required chips for sequencing. Multiomic cytometry involves cell staining with antibody-derived tags to identify targeted cellular surface epitopes. Consequently, cells were first labeled with cocktails of hashtag oligos (TotalSeq A0251 anti-human hashtag1 and TotalSeq A0252 anti-human hashtag2; BioLegend) and antibody-derived tag antibodies (TotalSeq A0126 anti-human CD133 and TotalSeq A0054 anti-human CD34; Bio-Legend) used at 1 M/ml as suggested, after saturation of non-specific sites with Human TruStain FCX FC Blocking Reagent (BioLegend). Then, around 20,000 labeled cells per sample were loaded into the microfluidic device to obtain between 5,000 and 10,000 encapsulated cells in GEMs. Barcoded cDNAs were amplified

and sent to the GENOM'IC platform (Cochin) for library preparation and sequencing on NextSeq 500/550 (Illumina).

## Bioinformatics analysis

### Time-lapse analysis

Time-lapse images were analyzed using *ImageJ* software (Rasband, W.S., *ImageJ*, U. S. National Institutes of Health, Bethesda, Maryland, USA; https://imagej.nih.gov/ij/). For proliferation and viability, cell tracking was performed manually using the *ImageJ* "Cell Counter" plugin. For each frame, living and dead cells were counted and events like cell divisions were recorded. The count tables were then exported in a .csv format to be graphically analyzed using custom R scripts (v3.6.0) with the ggplot2 package (v3.2.1). More than 150 microwells with one to eight initial cells were analyzed. For morphology tracking, a semi-automatic method in three steps was developed. First, images were automatically segmented with deep learning–based CellPose algorithm ([61]) and errors were removed with a homemade macro built on *ImageJ*. Then, cell tracking was performed with the Lineage Mapper *ImageJ* plugin. 50–100 cells were successfully tracked per condition. A unique identifier was attributed to individual cells based on segmented images and conserved along the cell cycle. At each division, the identifier was renewed with two others for daughter cells. This method allows for cell migration and division tracking and measuring cell cycle duration. Cell features over time (coordinates, area, perimeter…) were exported in a .csv file using another custom *ImageJ* macro. Finally, the table was used in R (v4.2.1) to compute roundness, cell cycle duration, and morphological switch rate (as described in Reference 9) and to generate plots with the ggplot2 package.

### Cytometry analysis

Flow cytometry .fsc files were analyzed with Kaluza software (v2.1). For short-term culture experiments, emission wavelengths of the chosen fluorochromes were sufficiently distant to not require a compensation step. Gates were manually adjusted using positive and negative controls, and generations were identified based on the CTV fluorescence intensity histogram. Proliferation indexes were calculated as described in Reference 62 with the following formula: $\frac{\sum_0^i N_i}{\sum_0^i \frac{N_i}{2^i}}$, where i is the generation number, and Ni is the number of events in generation i. For long-term culture experiments, a compensation matrix was automatically created using the monolabeled samples from all experiments at all time points. Manual checking of the obtained matrix and adjustments were next performed. Then, gates were placed using FMO data and unlabeled samples from each experiment. One gating protocol was established for each time point. Finally, compensation matrix and gating protocols were applied to the samples and statistical tables for each marker and combinations of markers were exported in a .csv format to be graphically plotted with R software (v4.3.0) and the ggplot2 package (v3.4.2).

### Mass spectrometry analysis

Results from the nine conducted experiments were merged into a common table using R (v4.3.1). Data were then normalized by the initial number of cells used for the experiments before the metabolite extraction. Samples showing excessive variations between

technical replicates were excluded from the dataset. For each metabolite, the mean values of the technical replicates were used for further analysis. The values obtained in each experimental condition were normalized to their respective controls.

### Bulk ATAC-seq analysis

Raw sequence data were cleaned and aligned on hg19 assembly by the CNRGH bioinformatics facility using common tools such as trimmomatic (v0.32), bowtie2 (v2.4.1), samtools (v1.4), macs2 (v2.1.1), picard (v1.138), and bedtools (v.2.29.2). Output .bam files were retrieved and handled with a custom snakemake (v7.25.0) workflow. First, the three biological replicates of each time point and condition were pooled together with the samtools merge function (v1.11). Other samtools functions were then used to index the files, extract quality control values, and count read numbers. No sample needed to be excluded based on quality control statistics. Using the samtools view function, we performed downsampling to the smallest number of reads detected in the cohort (80,000 000 reads). A new peak calling was conducted on downsampled .bam files using the macs2 callpeak tool (v2.2.7.1) with the following parameters: <-f BAMPE -g hs -B –broad –broad-cutoff 0.1>. Files were used as inputs in custom R scripts (v4.2.1). Readcount matrices were created for each merged sample with the featureCounts function (Rsubread v2.4.0), and a threshold on the minimum number of reads per peak was applied to remove technical background noise. Only peaks with 10 or more reads were kept for further analysis. Finally, Granges objects were created with the GenomicRanges package (v1.42.0) to be used in downstream analysis. To compare the accessibility profiles between different conditions and time points, a common list of regions was defined with Reduce:union function from GenomicRanges. Differential accessibility analysis was then performed following DESeq2 documentation (v1.38.3), and PCA was carried out with the prcomp function from the stats base package. Plots were drawn with ggplot2 (v3.4.2).

### Single-cell ATAC-seq analysis

Fastq files generated as sequencing output were cleaned and aligned to the hg38 reference genome by the bioinformatics facility of IMAGINE Institute using Cell Ranger software (10X Genomics, pipeline version: cellranger-atac-2.0.0). Count matrices were imported into R (v4.2.1) and integrated into a Chromatin Assay object using Seurat (v4.1.3) and Signac (v1.9.0) packages. Following the vignette's packages, QC metrics were computed to assess data quality and filter out peaks and cells of bad quality. Only cells with at least 200 peaks detected, nucleosome signal <2, blacklist ratio <0.05, TSS enrichment >2, number of fragments overlapping peaks between 3,000 and 50,000, and minimum 50% of total fragments fallen within ATAC-seq peaks were kept. Moreover, only peaks detected in at least 100 cells were retained and non-standard chromosomes were removed from the analysis. To allow comparison between the different conditions, a common set of peaks across the whole dataset was extracted with the reduce function from the Signac package and Chromatin Assay objects were merged. TF-IDF normalization followed by singular value decomposition was performed on the top features shared by more than 90% of the cells. Dimensions 2–50 of LSI reduction were used for dimensional reduction and clustering analysis. The Louvain

algorithm was used for clustering with a resolution 0.25. As the total cell counts differ across conditions, a normalization procedure was required to be able to compare the cells' distribution in clusters across conditions. Cells' numbers in each cluster were divided per the initial number of cells of the concerned condition and multiplied by 1,000. Gene ontology analysis was performed using org.Hs.eg.db (v3.16.0) and clusterProfiler (v4.6.2) packages. Most of the plots were drawn with the ggplot2 (v3.4.2) package.

### Comparison between bulk and single-cell ATAC-seq

To compare the bulk ATAC-seq and the scATAC-seq samples at 24 h, aligned reads from single-cell experiment were shifted from the hg38 to hg19 reference genome as described in Reference [1].

### Single-cell RNA-seq analysis

Demultiplexing and raw data alignment on the hg38 genome were conducted by the bioinformatics facility of GENOM'IC platform with Cell Ranger software (10X Genomics, pipeline version: cellranger-6.0.1, *cellranger multi* function). Seurat Objects were created from Cell Ranger count matrices with the Seurat package (v4.3.0) for downstream analysis on R (v4.2.1). Cumulative quality control filters were applied to the dataset to keep only good quality information. Inasmuch as the GEM formation is not free from a certain error's percentage, a filter on the minimum and maximum number of genes detected in a cell to be considered as a real one was applied to remove empty GEMs and doublets. Only cells with 1,000–6,500 detected genes were kept for further analysis. Moreover, cells presenting a high percentage of mitochondrial RNA and a low percentage of ribosomal RNA are either dead or dying. Thus, another filter was applied to keep cells with less than 25% of mRNA and more than 5% of rRNA. The 13 mitochondrial genes were also removed from the count matrices. Finally, only genes detected in more than three cells (among the 5,000 analyzed) were retained. After this cleaning step, normalization of the RNA assay was performed with the SCTransform algorithm (package sctransform v0.3.5) with *variable.features.n* and *ncells* parameters set to NULL and threshold on residual variance at 1.3. Meanwhile, the ADT assay from CITE-seq measurement was normalized across cells with the centered log-ratio method as advised on a Seurat multimodal data vignette. After confirmation that batch effect correction was not required, data from all conditions and experiments (control, inhibitors, and rescue) were combined into a single Seurat Object. Control samples from multiple experiments were shuffled and further considered as one. Linear dimensional reduction with PCA was then performed, and dimensions 1–40 were kept for later UMAP analysis and cluster identification. Using the SLM algorithm ([63]) and with the help of the clustree package (v0.5.0), cluster resolution 0.6 was selected leading to 17 cell clusters. As for the scATAC-seq analysis, cell's numbers were normalized by the total number of cells per condition. Downstream analysis was carried out using ggplot2 (v3.4.2) and Seurat functions for most of the plots. The FindAllMarkers function from Seurat was used to extract the top 20 positive markers for each cluster. Thresholds on adjusted *P*-value (<0.05) and percentage of cells expressing the markers (pct.1 – pct.2 > 0.25) were applied to the list before creating heatmap with the DoHeatmap function. Membrane protein groups of expression based on CITE-seq

results were defined using the kmeans function from the stats base package. Gene ontology analysis on differentially expressed genes between conditions was performed with org.Hs.eg.db (v3.15.0) and clusterProfiler (v4.4.4) packages. Two methods were used to extract lists of differentially expressed genes. On the one hand, visual outliers were identified on scatter plots of average gene expression. For each gene, an average UMI number of both the control and perturbed cells were calculated and reported on the x- and y-axis of the plot. Arbitrary thresholds at ±0.5 were set, highlighting genes that exhibit high responses (up-regulation or down-regulation) to the metabolic perturbation. On the other hand, another list of positive and negative markers was identified with the FindMarkers function from the Seurat package (test.use = "wilcox," logfc.threshold = 0.25, $P$_val_adj < 0.05). Both lists were used for gene ontology analysis. The pathway analysis was performed with ReactomeGSA (v1.10.0) on normalized gene expression to identify pathway-level enrichment between conditions. Before it, a list of interesting pathways had been extracted from the ReactomeContentService4R (v1.2.0) database.

## Data Availability

Data are available under the NCBI GEO accession number GSE243006. This accession number regroups three subseries of data: GSE243003 with bulk ATAC-seq data, GSE243004 with scATAC-seq data, and GSE243005 with scRNA-seq and CITE-seq data.

## Supplementary Information

## Acknowledgements

The authors are grateful to Romain Morichon, Daniel Stockholm, and Jérémie Cosette for their help with time-lapse recording, to Peggy Sanatine for providing training in the use of the cytometer SP6800 for CTV cytometry, and to Thierry Jaffredo for his support during the use of the Chromium 10X device for scRNA-seq experiment. We also acknowledge the use of instruments and resources at the mass spectrometry facility at inStem and NCBS. The authors would like to extend special thanks to Camille Baey from BioLegend for her assistance, valuable advice, and time dedicated to the setup and optimization of the long-term cytometry panel. Moreover, we would like to thank all the members of the platforms that performed sequencing and bio-informatics preprocessing of the NGS data: Brigitte Izac and Benjamin SaintPierre from the GENOM'IC platform, Marine Luka and Francesco Carbone from IMAGINE platform, and Sophie Chantalat from CNRGH. The authors are also grateful to the mothers and the staff of the Centre Hospitalier Sud Francilien and AP-HP Hôpital Saint-Louis for umbilical cord blood samples. We are grateful to Prof. Takuya Imamura for his valuable comments and suggestions. This work was supported by the EPHE grant (11REC/BIMO to A Paldi) and ANR grant (ANR-17CE12-0031-01 «SinCity » to A Paldi). The funders had no role in study design, data collection and analysis, decision to publish, or preparation of the article.

## Author Contributions

L Racine: data curation, software, formal analysis, validation, investigation, visualization, methodology, and writing—original draft, review, and editing.
R Parmentier: data curation, software, formal analysis, investigation, visualization, and methodology.
S Niphadkar: data curation, formal analysis, and investigation.
J Chhun: software, investigation, and methodology.
J-A Martignoles: data curation, formal analysis, and validation.
F Delhommeau: formal analysis, funding acquisition, and validation.
S Laxman: conceptualization, supervision, validation, methodology, and writing—original draft.
A Paldi: conceptualization, data curation, formal analysis, supervision, funding acquisition, validation, visualization, methodology, and writing—original draft, review, and editing.

## Conflict of Interest Statement

The authors declare that they have no conflict of interest.

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
