## [Reviewer comments · Life Science Alliance]

Life Science Alliance

Metabolic adaptation pilots the differentiation of human hematopoietic cells.

Laetitia Racine, Romuald Parmentier, Shreyas Niphadkar, Julie Chhun, Jean-Alain Martignoles, Francois Delhommeau, Sunil Laxman, and Andras Paldi

DOI: <https://doi.org/10.26508/lsa.202402747>

Corresponding author(s): Andras Paldi, *École Pratique des Hautes Études*

Review Timeline:	Submission Date:	2024-03-29
	Editorial Decision:	2024-05-08
	Revision Received:	2024-05-12
	Accepted:	2024-05-13

Transaction Report:

Please note that the manuscript was reviewed at *Review Commons* and these reports were taken into account in the decision-making process at *Life Science Alliance*.

Review
COMMONS

Reviews

Review #1

The current study investigates the metabolic regulation of hematopoietic cell differentiation through chromatin modification and gene expression. Using the primary CD34+ human cord blood cells, the authors show that transient pharmacological inhibition of glycolysis, PPP, and glutamine/glutamate metabolism alters the dynamics of chromatin structures and gene expression, leading to the impacts on cell proliferation, morphology, and the long-term differentiation capacity. Following are specific comments:

****Major:****

1. The rationale behind the selection of the metabolic targets and the working hypothesis regarding specific effects on cellular consequence is not explicitly conveyed, which makes it difficult to judge if the experiment design is appropriate and if the results address the questions:
 - i. The operational definition of "Metabolic perturbation" or "Metabolic stress" needs to be provided and the validation of inhibitory effects needs to be clarified. Fig. 3D and S1 Fig are supposed to indicate the inhibition of targeted metabolic pathways but it is not clear if the authors believe the inhibitors exert expected metabolic effects based on the presented data. The author should explain why they target the selected pathways (i.e. glycolysis, PPP and glutamine/glutamate metabolism) and precisely point out which up or down regulation (in Fig. 3D and S1 Fig, for example) indicate sufficient and specific inhibitory effects for each inhibitor to operationally define "metabolic perturbation".
 - ii. Given that the major goal of the study is to characterize the long-term effects of transient metabolic perturbation, it is particular important to address how soon after the treatment (and how soon after removal) of the inhibitor, the authors observed the expected changes of the targeted metabolic pathways.
2. The chromatin-independent and transcriptional-independent mechanisms are not considered. Intermediate metabolites are known to directly modify protein activity, alter cell signaling resulting changes in differentiation potentials. The authors should acknowledge this possibility and examining their data to speculate which specific gene expression and related cell-fate changes are likely (or not likely) the direct result of epigenetic modulation.
3. The samples of primary cells have heterogenic cell populations. The cellular characterization in bulk may confound the results regarding cell-fate programming versus the cell selection effect.
 - i. In Fig 3 and Fig6, how would the authors determine whether the inhibitor or rescue treatments alter cell differentiation program or selectively allow proliferation or survival of non-differentiated cells?
 - ii. Trajectory analysis may further elucidate that the effects of metabolic perturbation on cell differentiation program are permissive or more instructive (towards/against specific lineage commitment).

****Minor:****

1. Fig. 1A is missing figure legends.
2. The cell clusters in fig 3 needs to be at least deconvoluted based on the differentiation or cell-identity markers and annotated accordingly in the main figure.
3. The statements in abstract and introduction broadly mention the environmental changes and metabolic adaptation in the process of differentiation. The study, however, address only the setting in vitro. As the mobilization of the hematopoiesis process is not possible to be address with the data presented in the current study. The author should revise the manuscript to better introduce relevant questions of the study.

Overall, we appreciate the author using untrivial experiments with purified/primary human cells and highly parallel omics analyses to test an interesting hypothesis. However, we think the specific question(s) and objective(s) of the study need to be specified/clarified and to be better addressed by more conclusive results.

This study will be of fundamental interest to the field of stem cell biology, cell metabolism and developmental biology. Our expertise is adult stem cell biology and dietary research.

Review #2

****Summary:****

The authors evaluate the impact of metabolic perturbations on chromatin structure and the transcriptional landscape of undifferentiated hematopoietic progenitor cells following stimulation with early acting cytokines. Of

note, the authors find very early changes in chromatin structure, associated with more long-term changes in transcriptional profiles, modulating the differentiation potential of these progenitors.

****Major Comments:****

- The authors show a significantly larger impact of AOA than DON on the chromatin and transcription responses of CD34+ progenitors even though they are both impacting glutamine metabolism. Alpha-ketoglutarate rescued CD34+ progenitors from the effect of AOA but did not rescue DON-treated cells which should also have an attenuated generation of alpha-ketoglutarate. How do the authors interpret this apparent discrepancy? In this regard, the MS data are confusing to this reviewer; alpha ketoglutarate levels were much higher in AOA-treated cells than in DON (or even 2-DG-treated) cells, potentially suggesting that DON had more of an impact on glutamine metabolism than AOA. Additionally, glutamine levels are low in DON-treated cells (where GLS is inhibited) but not in AOA-treated cells (this reviewer would have expected higher levels in both) and lactate is high in 2-DG treated cells (low levels would have been expected).
- The authors' finding of a single cluster of cells following AOA treatment (cluster 8) is extremely impressive. Can the authors better define this cluster?
- The authors find an increase in cells expressing the CD36 marker, especially following 2-DG treatment. However, they never discuss the functional significance of CD36 as a fatty acid translocase (FAT), serving as a receptor for long chain fatty acids, and potentially as a compensatory mechanism under conditions where glucose metabolism is inhibited.

****Minor Comments:****

- A schematic showing the different inhibitors and metabolic pathways would be helpful.

General comments:

The impact of metabolic perturbations on a progenitor cell with the potential to differentiate to multiple lineages is of much interest to the field. The authors have performed extensive single cell analyses, incorporating both scATACseq and scRNAseq together with cell morphology analyses and cell surface protein evaluations, to monitor short and long term impacts. They find very rapid changes in chromatin structure with long-lasting effects, despite the cessation of the metabolic perturbation. This has important implications for our understanding of the crosstalk between metabolic alterations, chromatin structure, and gene expression, coming together to regulate progenitor cell survival, expansion, and differentiation.

Assessments: strengths and limitations

Strengths and Advances: The authors should be commended for their use of primary hematopoietic progenitors and a close evaluation of the impact of metabolic perturbations during the first 24h of stimulation. Their studies have added significantly to our understanding of cell differentiation, showing that changes in metabolic circuits rapidly modulate cytokine-induced epigenetic chromatin states.

Limitations: Because CD34+ progenitors represent a heterogeneous population, metabolic perturbations are likely impacting the different subsets in distinct manners. The single cell data presented here can be exploited to assess how these subsets (clusters) change at very early time points following perturbation. It will also be important to confirm the effects of different inhibitors on specific metabolites in a cell line(s) since the changes reported here do not appear to be specific. It is possible that these differences are due to an overall decrease in the activation state of a cytokine-stimulated progenitor leading to a global decrease in metabolites.

Audience:

This study will be of much interest to scientists/clinicians studying stem cells, hematopoietic stem cells, metabolism, and epigenomic/transcriptomic landscapes. As such, it will be of interest to a large community.

1. General Statements [optional]

The authors thank the reviewers for their comments and suggestions that helped to improve the manuscript.

This section is mandatory. Please insert a point-by-point reply describing the revisions that were already carried out and included in the transferred manuscript.

Reviewer #1 (Evidence, reproducibility and clarity (Required)):
--

Summary:

The current study investigates the metabolic regulation of hematopoietic cell differentiation through chromatin modification and gene expression. Using the primary CD34+ human cord blood cells, the authors show that transient pharmacological inhibition of glycolysis, PPP, and glutamine/glutamate metabolism alters the dynamics of chromatin structures and gene expression, leading to the impacts on cell proliferation, morphology, and the long-term differentiation capacity. Following are specific comments:

Major:

1. The rationale behind the selection of the metabolic targets and the working hypothesis regarding specific effects on cellular consequence is not explicitly conveyed, which makes it difficult to judge if the experiment design is appropriate and if the results address the questions:

i. The operational definition of "Metabolic perturbation" or "Metabolic stress" needs to be provided and the validation of inhibitory effects needs to be clarified. Fig. 3D and S1 Fig are supposed to indicate the inhibition of targeted metabolic pathways but it is not clear if the authors believe the inhibitors exert expected metabolic effects based on the presented data. The author should explain why they target the selected pathways (i.e. glycolysis, PPP and glutamine/glutamate metabolism) and precisely point out which up or down regulation (in Fig. 3D and S1 Fig, for example) indicate sufficient and specific inhibitory effects for each inhibitor to operationally define "metabolic perturbation".

Thank you for bringing this point to our attention. We extended the Introduction section (page 3) with a paragraph better explaining the notion of metabolic perturbation or stress. Indeed, a clear definition of the metabolic targets is also required. Consequently, the update includes a more detailed presentation of the metabolic steps and the rationale as to why we selected them

as targets (pages 3 to 4). Additionally, we have also incorporated an extra figure (SI Fig) to illustrate the major metabolic pathways affected by the various inhibitors.

In this study, we have used single time-point detections of steady-state metabolite levels. The single time-point detection of individual metabolite levels alone does not allow clear understanding of the precise metabolic alterations. The network of metabolic reactions is highly interconnected with complex regulatory loops that makes precise predictions difficult. More detailed metabolic flux studies will be required to characterize the perturbations. There are considerable challenges in carrying out such flux experiments with the limited amount of cells (which cannot all be from a single patient source), making such experiments well beyond the scope of this study. However, even with single time-point steady inhibitor studies, we observe significant and inhibitor-specific cellular reactions involving cell division rate, morphology, cell surface marker distribution and changes in bulk metabolite levels. Therefore, we interpret these changes as collectively reflecting the metabolic impact of the inhibitors, which can be qualified as metabolic perturbation or stress. The manuscript has been modified (page 5) to clarify this point.

ii. Given that the major goal of the study is to characterize the long-term effects of transient metabolic perturbation, it is particularly important to address how soon after the treatment (and how soon after removal) of the inhibitor, the authors observed the expected changes of the targeted metabolic pathways.

The cells were cultured in the presence of inhibitors for 4 days, with day 0 being the beginning of the experiment. The effect on chromatin was detectable by ATAC-seq as early as 12 hours. Given the dramatic changes observed at 24h and early changes (detected at the chromatin level and observed in Time-Lapse), it is reasonable to infer that changes occur almost immediately after the addition of the inhibitors. The first time point that was analyzed after the removal of inhibitors was on day 7 (i.e. 3 days culture without inhibitors), then on day 10 and 14. The cells of the four conditions exhibited distinct evolution even after the inhibitors were removed.

2. The chromatin-independent and transcriptional-independent mechanisms are not considered. Intermediate metabolites are known to directly modify protein activity, alter cell signaling resulting changes in differentiation potentials. The authors should acknowledge this possibility and examining their data to speculate which specific gene expression and related cell-fate changes are likely (or not likely) the direct result of epigenetic modulation.

We completely agree with the reviewer that cellular memory mechanisms other than chromatin modifications were not investigated. Fluctuations of the energy metabolism can also impact the post-translational modifications of cellular proteins. However almost nothing is known so far on the role of these modifications in cellular memory processes, and in the consolidation of phenotypic characteristics of a cell lineage. This idea is of course very exciting, but studying this aspect would necessitate an entirely separate investigation, using alternative methods. At this stage we believe that this is well out of the scope of the present study. We have added the idea in the Discussion section (page 16).

3. The samples of primary cells have heterogenic cell populations. The cellular characterization in bulk may confound the results regarding cell-fate programming versus the cell selection effect.

i. In Fig 3 and Fig6, how would the authors determine whether the inhibitor or rescue treatments alter cell differentiation program or selectively allow proliferation or survival of non-differentiated cells?

The question of the first selective hit followed by the amplification of the surviving cells is highly relevant. The CD34⁺ cell population is inherently very heterogenous, and we used inhibitor concentrations close to the IC50 values. Collectively, we observe that the surviving cells exhibited greater resistance, which is likely due to their more resistant metabolic state. Our metabolic MS analysis was conducted on a bulk population, precluding conclusions at the single-cell level. However, time-lapse, cytometry, single-cell ATAC and RNA-seq analyses all provide information at the single-cell level. ATAC-seq revealed initial differences between control and treated cells approximately 12 hours after stimulation. By 24 hours, 16 different subsets of cells were identified using single-cell ATAC-seq chromatin accessibility profiling. All four conditions were represented in all subsets in variable proportions. Previous studies [1,9] indicated that at 24 hours, these cells couldn't be clustered into distinct groups based on their gene expression patterns, suggesting that chromatin changes precede gene expression changes by several hours. Notably, at the time of analysis, these cells had not undergone division yet. Time-lapse microscopy revealed that the first division occurred in control and 2-DG cells 24 hours later, while in DON and AOA cells, it occurred only around 72 hours later. At this point, single-cell RNA-seq data clustering identified 17 different subsets of cells. Particularly, AOA cells exhibited a distinctly different gene expression pattern, forming separate clusters. Based on these observations, we think that although some selection occurs during the initial hours, the differences observed between the inhibitors cannot be solely explained by it. Instead, chromatin differences between cells appear before the first division of the cells surviving the initial shock. These differences then gradually develop over the initial 96 hours. The inhibitors were removed at this point, and the cells primed by the different inhibitors were subsequently cultured under identical conditions. It is likely that cells exhibiting differential gene expression patterns possessed varying proliferation capacities, contributing to the observed evolution of cell populations as detected on days 7, 10, and 14. We have added this paragraph to the manuscript in the Discussion section for better clarity (pages 14 and 15).

ii. Trajectory analysis may further elucidate that the effects of metabolic perturbation on cell differentiation program are permissive or more instructive (towards/against specific lineage commitment).

Although we were able to identify 17 subsets of cells based on their transcriptome profiles, any of them could be assigned to a specific hematopoietic lineage. It is presumably too early. As it was shown (Moussy et al 2017), at this stage, just 96 hours after stimulation most of the cells are still “hesitant” with fluctuating gene expression profiles and morphology. Their commitment to a specific lineage is not robust making the definition of trajectories impossible.

Minor:

1. Fig. 1A is missing figure legends.

We clarified the legend (see page 40).

2. The cell clusters in fig 3 needs to be at least deconvoluted based on the differentiation or cell-identity markers and annotated accordingly in the main figure.

Indeed, we conducted this analysis, but the results weren't conclusive enough to be included in the manuscript. We extracted the list of differentially expressed genes for each cluster (for a more detailed description, refer to the answer to Reviewer 2's Question 2 regarding the analysis of cluster 8). The list of extracted biomarkers was studied, and the top 20 for each cluster are shown on the heat-map in S6 Fig. However, for many clusters, canonical markers couldn't be identified to easily match the clusters to known cell types. For others, a few markers were

detected, but with inconsistent mixes, such as in cluster 7 (LYZ and CD14 associated with CD14+ Mono, CST3 associated with DC, NKG7 associated with NK, IL7R and S100A4 associated with Memory CD4+, and MS4A7 associated with B cells) or in cluster 12 (PPBP associated with platelets, S100A4 associated with memory CD4+ cells and FCER1A associated with DC). At this very early stage, the cells are just exiting the multi-lineage primed stage, and it's likely that their identity is not yet fully determined, explaining the mix of markers from different lineages. We also attempted a Gene Ontology analysis on the lists of biomarkers, but most terms were general cellular functioning terms, making it impossible to assign the cells in the various clusters to specific cell types.

3. The statements in abstract and introduction broadly mention the environmental changes and metabolic adaptation in the process of differentiation. The study, however, address only the setting in vitro. As the mobilization of the hematopoiesis process is not possible to be address with the data presented in the current study. The author should revise the manuscript to better introduce relevant questions of the study.

With all due respect, we do not agree with this comment. The question we are seeking the answer to is defined in the Introduction section (page 3): “Does the change of the metabolic setup of the cells precede and trigger the non-specific chromatin opening?”. For better clarity, now we extended this question by a second one (page 3). It is true that in vitro studies cannot reproduce faithfully all the in vivo conditions such as the mobilization of the hematopoiesis process. However, the objective of our study was only to ask if the external restriction of the energy metabolism modifies the cellular differentiation process. From this perspective, utilizing metabolic inhibitors is a possible way to model restricted access to some substrates in a stressful environment. Indeed, this is the entire philosophy and value of in vitro experiments. The time resolution used in this study is impossible to achieve currently in any in vivo setting. The use of human CD34⁺ cells was motivated by the fact that this is a very well-studied in vitro model that retains many characteristics of cell differentiation in general. We only hope that our hypothesis and the observations done here are robust enough to be generalizable to other models and to cell differentiation in general. Obviously, confirmation by complementary studies on various other cellular models will be required.

Reviewer #1 (Significance (Required)):

Overall, we appreciate the author using untrivial experiments with purified/primary human cells and highly parallel omics analyses to test an interesting hypothesis. However, we think the specific question(s) and objective(s) of the study need to be specified/clarified and to be better addressed by more conclusive results.

This study will be of fundamental interest to the field of stem cell biology, cell metabolism and developmental biology. Our expertise is adult stem cell biology and dietary research.

Reviewer #2 (Evidence, reproducibility and clarity (Required)):

Summary:

The authors evaluate the impact of metabolic perturbations on chromatin structure and the transcriptional landscape of undifferentiated hematopoietic progenitor cells following stimulation with early acting cytokines. Of note, the authors find very early changes in chromatin structure, associated with more long-term changes in transcriptional profiles, modulating the differentiation potential of these progenitors.

Major Comments:

-The authors show a significantly larger impact of AOA than DON on the chromatin and transcription responses of CD34+ progenitors even though they are both impacting glutamine metabolism. Alpha-ketoglutarate rescued CD34+ progenitors from the effect of AOA but did not rescue DON-treated cells which should also have an attenuated generation of alpha-ketoglutarate. How do the authors interpret this apparent discrepancy? In this regard, the MS data are confusing to this reviewer; alpha ketoglutarate levels were much higher in AOA-treated cells than in DON (or even 2-DG-treated) cells, potentially suggesting that DON had more of an impact on glutamine metabolism than AOA. Additionally, glutamine levels are low in DON-treated cells (where GLS is inhibited) but not in AOA-treated cells (this reviewer would have expected higher levels in both) and lactate is high in 2-DG treated cells (low levels would have been expected).

We were surprised by the metabolite levels found by mass spectrometry in the cells at 24 hours. In many cases these levels were different than what one would intuitively expect. This is why we have repeated the experiments many times. One possible explanation is to consider that these metabolites are produced and consumed simultaneously by many different alternative biochemical reactions. Inhibiting one of them induces immediate compensations by others. The metabolic network is complex and its state at a given moment is difficult to predict. Our measurement provides only a snapshot (which are steady-state measurements at that time). The significant change in the abundance of many metabolic intermediates indicates the fact that the network function is perturbed. To understand in detail the exact nature of these perturbations a single time-point measurement is not sufficient, detailed metabolic flux studies will be able to identify the modified metabolic fluxes. This is at present challenging, because the sources of cells are from different patients, at different times, and will require overcoming substantial experimental challenges. More specifically, the reason why AOA had a greater impact on the chromatin than DON and could be rescued by alpha-ketoglutarate may reside in the structure of the glutamine metabolizing pathway. The effect of DON inhibition on alpha-ketoglutarate can be relatively easily compensated by other amino acids, given that glutamine is a non-essential amino acid. This aligns with the observed recovery of surviving cells after an initial setback, where they subsequently resume their proliferation and differentiation following a brief lag period. Conversely, compensating for the inhibition caused by AOA is more challenging due to the direct involvement of transaminases in α KG production.

The manuscript has been completed in the Results section (page 5) and in the Discussion section (pages 15 and 16).

-The authors' finding of a single cluster of cells following AOA treatment (cluster 8) is extremely impressive. Can the authors better define this cluster?

Indeed, scRNA-seq analysis at 96hrs revealed very specific transcriptomic profiles for the AOA condition (Fig.3BC). Although some cells appeared in small numbers in clusters common to other conditions (clusters 4, 7, 10 and 13), most were grouped in completely distinct clusters (clusters 8, 11, 14 and 15). In particular, cluster 8 contained 70.2% of the cells from the AOA condition, i.e. 3598 cells out of 5126 analyzed for this condition before normalization. Given the small size of clusters 11, 14 and 15, attention was focused on cluster 8 for further characterization.

First, we were able to confirm that this cluster was real and significant because even at a lower resolution than that initially used for the study (resolution 0.6 in Fig.3B), the cluster persists, so it is not an artefact of the clustering algorithm (cluster 1 on the figure on the left corresponds to cluster 8 on Fig.3B).

Overall, the analysis of gene expression profile revealed that the cluster 8 was better defined by the genes that were down regulated rather than those overexpressed compared to the other clusters. However, the Gene Ontology analysis conducted on these gene lists was inconclusive. The extracted biomarkers do not allow for associating the cells with a specific mature cell type, 96hours is too early in the differentiation process. We think that this observation is not sufficiently conclusive at this stage to be included in the manuscript. Deeper analyses would be necessary to better understand their specificity, but it was out of the scope of the present study.

Here is the detailed description of the analysis:

We searched for specific markers to characterize this cluster using the FindAllMarkers function in the Seurat package. This analysis compares each cluster against all others, identifying genes with differential expression. In the generated output, pct.1 represents the proportion of cells within the cluster where a specific gene is detected, while pct.2 signifies the average proportion of cells across all other clusters where the gene is detected. To refine our results, we filter the positive markers, retaining those with a difference > 0.25 between pct.1 and pct.2, alongside a p_val_adj < 0.05. Employing this approach, we derived a list of 14 genes overexpressed in cluster 8: CRHBP, PRKACB, PNRC1, SNHG32, NRIP1, GPIBB, FAM117A, ANKRD28, GBP4, GCSAML, BEX2, NPR3, CASC15, and SPINK2. Subsequently, we conducted a Gene Ontology analysis on this gene list using the ClusterProfiler package. However, only three terms emerged from this analysis:

ID	Ont.	Description	Gene Ratio	geneID	Count
GO:0071392	BP	cellular response to estradiol stimulus	45171	CRHBP/NRIP1	2
GO:0017046	MF	peptide hormone binding	45232	CRHBP/NPR3	2

GO:0042562	MF	hormone binding	45232	CRHBP/NPR3	2
------------	----	-----------------	-------	------------	---

The study of genes overexpressed in this cluster 8 is therefore inconclusive. When we look at the heatmap with the top 20 markers for each cluster, it seems that cluster 8 is characterized by the under-expression of certain genes, genes that are also under-expressed in clusters 14 and 15 and over-expressed in clusters 11 and 16: GPNMB, LGALS3, MMP9, CTSD, CXCL8, CTSB, SOD2, IFI30, PSAP, CHI3L1, CYP1B1, CSTB, ACP5, MARCKS, S100A11, FCER1G, LIPA. We conducted a Gene Ontology analysis on this new list, and this time, 53 terms were identified. The figure below shows the top 25 terms. Several terms related to immune cells and neutrophils are observed. The standard analysis doesn't provide us with additional insights into the cells within cluster 8.

-The authors find an increase in cells expressing the CD36 marker, especially following 2-DG treatment. However, they never discuss the functional significance of CD36 as a fatty acid translocase (FAT), serving as a receptor for long chain fatty acids, and potentially as a compensatory mechanism under conditions where glucose metabolism is inhibited. *We thank the reviewer for drawing our attention to this omission. It is indeed highly relevant and important to mention it in the paper. It fits perfectly with the basic idea of metabolic adaptation as a driving force. We introduced this point with references in the manuscript in the Results section (page 11).*

Minor Comments:

-A schematic showing the different inhibitors and metabolic pathways would be helpful. *A schematic representation of the main metabolic pathways and the steps affected by inhibitors has been added as S1 Fig (see page 32 and 40). Consequently, the other supplementary figures have been renumbered.*

Reviewer #2 (Significance (Required)):

General comments:

The impact of metabolic perturbations on a progenitor cell with the potential to differentiate to multiple lineages is of much interest to the field. The authors have performed extensive single cell

analyses, incorporating both scATACseq and scRNAseq together with cell morphology analyses and cell surface protein evaluations, to monitor short- and long-term impacts. They find very rapid changes in chromatin structure with long-lasting effects, despite the cessation of the metabolic perturbation. This has important implications for our understanding of the crosstalk between metabolic alterations, chromatin structure, and gene expression, coming together to regulate progenitor cell survival, expansion, and differentiation.

Assessments: strengths and limitations

Strengths and Advances:

The authors should be commended for their use of primary hematopoietic progenitors and a close evaluation of the impact of metabolic perturbations during the first 24h of stimulation. Their studies have added significantly to our understanding of cell differentiation, showing that changes in metabolic circuits rapidly modulate cytokine-induced epigenetic chromatin states.

Limitations:

Because CD34+ progenitors represent a heterogeneous population, metabolic perturbations are likely impacting the different subsets in distinct manners. The single cell data presented here can be exploited to assess how these subsets (clusters) change at very early time points following perturbation. It will also be important to confirm the effects of different inhibitors on specific metabolites in a cell line(s) since the changes reported here do not appear to be specific. It is possible that these differences are due to an overall decrease in the activation state of a cytokine-stimulated progenitor leading to a global decrease in metabolites.

Audience:

This study will be of much interest to scientists/clinicians studying stem cells, hematopoietic stem cells, metabolism, and epigenomic/transcriptomic landscapes. As such, it will be of interest to a large community.

May 8, 2024

RE: Life Science Alliance Manuscript #LSA-2024-02747

Prof. Andras Paldi
Ecole Pratique des Hautes Etudes, PSL Research University
Centre de Recherche Saint Antoine
27, rue Chaligny
Paris 75012
France

Dear Dr. Paldi,

Thank you for submitting your revised manuscript entitled "Metabolic adaptation pilots the differentiation of human hematopoietic cells.". We would be happy to publish your paper in Life Science Alliance pending final revisions necessary to meet our formatting guidelines.

- please be sure that the authorship listing and order is correct
- please upload your main manuscript text as an editable doc file
- please upload all figure files as individual ones, including the supplementary figure files; all figure legends should only appear in the main manuscript file
- please add a Running Title and a Summary Blurb/Alternate Abstract to our system
- please add a Category for your manuscript in our system
- please add the Twitter handle of your host institute/organization as well as your own or/and one of the authors in our system
- please be sure to add the Author's Contributions to our system as well
- please add your main, supplementary figure, and table legends to the main manuscript text after the references section
- please separate the Figure legends and Supplemental Figure legends into separate sections
- please upload your Tables in editable .doc or Excel format
- we encourage you to revise the figure legends for Figure 3 such that the figure panels are introduced in alphabetical order
- please add callouts for Figures 4B and S7A-B to your main manuscript text

A. FINAL FILES:

B. MANUSCRIPT ORGANIZATION AND FORMATTING:

Sincerely,

Reviewer #2 (Comments to the Authors (Required)):

As indicated in my initial review, the authors incorporate both scATACseq and scRNAseq together with cell morphology analyses and cell surface protein evaluations, to monitor short- and long-term impacts of metabolic pathways on HSC differentiation potential. The authors detect very early changes in chromatin structure and these changes are coupled to more long-term changes in transcriptional profiles and progenitor differentiation potential. Thus, these data have important implications for our understanding of the crosstalk between metabolic alterations, chromatin structure, and gene expression, with long-term effects on hematopoietic progenitor cell differentiation.

The authors have appropriately responded to the comments/critiques that were raised.

May 13, 2024

RE: Life Science Alliance Manuscript #LSA-2024-02747R

Prof. Andras Paldi
École Pratique des Hautes Études
Centre de Recherche Saint Antoine
27, rue Chaligny
Paris 75012
France

Dear Dr. Paldi,

Thank you for submitting your Research Article entitled "Metabolic adaptation pilots the differentiation of human hematopoietic cells.". It is a pleasure to let you know that your manuscript is now accepted for publication in Life Science Alliance. Congratulations on this interesting work.

DISTRIBUTION OF MATERIALS:

Again, congratulations on a very nice paper. I hope you found the review process to be constructive and are pleased with how the manuscript was handled editorially. We look forward to future exciting submissions from your lab.

Sincerely,
